# Polθ inhibitors elicit *BRCA*-gene synthetic lethality and target PARP inhibitor resistance

Diana Zatreanu[1,2,10], Helen M. R. Robinson[3,10], Omar Alkhatib[3], Marie Boursier[3], Harry Finch[3], Lerin Geo[3], Diego Grande [3], Vera Grinkevich[3], Robert A. Heald[3], Sophie Langdon[3], Jayesh Majithiya[3], Claire McWhirter [3], Niall M. B. Martin[3], Shaun Moore[3], Joana Neves[3], Eeson Rajendra [3], Marco Ranzani[3], Theresia Schaedler[3], Martin Stockley[3], Kimberley Wiggins[3], Rachel Brough[1,2], Sandhya Sridhar[1,2], Aditi Gulati[1,2], Nan Shao[1,2], Luned M. Badder[4], Daniela Novo[2], Eleanor G. Knight[2], Rebecca Marlow[2,4], Syed Haider [2], Elsa Callen[5], Graeme Hewitt[6], Joost Schimmel[7], Remko Prevo[8], Christina Alli[9], Amanda Ferdinand[9], Cameron Bell[9], Peter Blencowe[9], Chris Bot [9], Mathew Calder[9], Mark Charles[9], Jayne Curry[9], Tennyson Ekwuru [9], Katherine Ewings[9], Wojciech Krajewski[9], Ellen MacDonald[9], Hollie McCarron[9], Leon Pang[9], Chris Pedder[9], Laurent Rigoreau[9], Martin Swarbrick [9], Ed Wheatley[9], Simon Willis[9], Ai Ching Wong[9], Andre Nussenzweig[5], Marcel Tijsterman [7], Andrew Tutt[2,4], Simon J. Boulton [3,6], Geoff S. Higgins[8], Stephen J. Pettitt [1,2✉], Graeme C. M. Smith [3✉] & Christopher J. Lord [1,2✉]

To identify approaches to target DNA repair vulnerabilities in cancer, we discovered nanomolar potent, selective, low molecular weight (MW), allosteric inhibitors of the polymerase function of DNA polymerase Polθ, including ART558. ART558 inhibits the major Polθ-mediated DNA repair process, Theta-Mediated End Joining, without targeting Non-Homologous End Joining. In addition, ART558 elicits DNA damage and synthetic lethality in *BRCA1*- or *BRCA2*-mutant tumour cells and enhances the effects of a PARP inhibitor. Genetic perturbation screening revealed that defects in the 53BP1/Shieldin complex, which cause PARP inhibitor resistance, result in in vitro and in vivo sensitivity to small molecule Polθ polymerase inhibitors. Mechanistically, ART558 increases biomarkers of single-stranded DNA and synthetic lethality in 53BP1-defective cells whilst the inhibition of DNA nucleases that promote end-resection reversed these effects, implicating these in the synthetic lethal mechanism-of-action. Taken together, these observations describe a drug class that elicits *BRCA*-gene synthetic lethality and PARP inhibitor synergy, as well as targeting a biomarker-defined mechanism of PARPi-resistance.

---

[1] CRUK Gene Function Laboratory, The Institute of Cancer Research, London, UK. [2] The Breast Cancer Now Toby Robins Research Centre, The Institute of Cancer Research, London, UK. [3] Artios Pharma, The Glenn Berge Building, Babraham Research Campus, Cambridge, UK. [4] The Breast Cancer Now Research Unit, King's College London, London, UK. [5] Laboratory of Genome Integrity, National Cancer Institute, NIH, Bethesda, MD, USA. [6] The Francis Crick Institute, London, UK. [7] Department of Human Genetics, Leiden University Medical Center, Leiden, The Netherlands. [8] Medical Research Council Oxford Institute for Radiation Oncology, University of Oxford, Old Road Campus Research Building, Roosevelt Drive, Oxford, UK. [9] Cancer Research UK, Therapeutic Discovery Laboratories, Jonas Webb Building, Babraham Research Campus, Cambridge, UK. [10]These authors contributed equally: Diana Zatreanu, Helen M. R. Robinson. ✉email: Stephen.Pettitt@icr.ac.uk; Graeme.Smith@artiospharma.com; Chris.Lord@icr.ac.uk

The repair of double-stranded DNA breaks (DSB) can be broadly classified into three main pathways; non-homologous end joining (NHEJ), which preferentially repairs unresected DSB ends[1–3] and two processes that require nucleolytic resection of 5′ terminal strands generating DSBs with a 3′ ssDNA overhang[4–6]. These latter processes are termed homologous recombination (HR), a conservative template-dependent DNA repair process requiring the BRCA1 and BRCA2 tumour suppressor proteins, and an error-prone process, theta-mediated end joining (TMEJ, also known as alt-NHEJ or microhomology-mediated end joining, MMEJ). HR is a largely error-free mechanism of DSB repair, which utilises stand invasion into an intact sister chromatid or homologous chromosome followed by templated DNA synthesis to repair the damage. In cells that lack HR, such as *BRCA*-gene deficient cancer cells, TMEJ serves as an essential backup pathway to repair resected DSBs[7]. TMEJ is initiated by 5′ to 3′ resection factors, involves the poly-(ADP-ribose) polymerase PARP1, DNA ligase III and the eponymous 290 kDa Polymerase A family enzyme, DNA polymerase theta (Polθ, encoded by *POLQ*)[8]. Polθ possesses a N-terminal helicase-like domain and a C-terminal DNA polymerase domain separated by a non-structured central amino acid sequence[9,10] and is only found in multicellular organisms, where it is relatively well-conserved[11]. The polymerase domain of Polθ includes three insertion amino acid loops, not conserved among other A-family DNA polymerases[9]. It is this distinct structure that allows for the interaction, annealing, and extension of short single-stranded (ss) DNA primers[12,13]. Biochemical studies have shown that the helicase domain of Polθ acts to displace RPA bound to the single-strand DNA overhang and facilitate annealing of short tracts of microhomology (>1-2 bp) that flank a DSB, potentially using distant DNA sites as templates[4,12,14,15]. Polθ then employs its polymerase domain to initiate DNA synthesis to fill in the gaps, prior to ligation of the annealed DSB ends.

The interest in Polθ as a therapeutic target in cancer has been piqued by a number of observations including synthetic lethal interactions between loss of the *POLQ* gene and deficiencies in DNA repair-related tumour suppressor genes that control DSB repair/HR, including *BRCA1, BRCA2, ATM* and *FANCD2*, observations perhaps best explained by the role TMEJ plays as a backup pathway in the absence of HR[7,13,16–18]. As for the vast majority of cancer-related synthetic lethal effects identified by genetic perturbation, the potential to exploit *POLQ*/HR-gene synthetic lethal effects have not as yet been realized by the discovery of small molecule inhibitors[7]. In part at least, this might be due to the perceived complexity in identifying potent and selective inhibitors of DNA polymerases or helicases, as opposed to other drug targets such as protein kinases.

Here, we describe the discovery of ART558, a small molecule inhibitor of the DNA polymerase activity of Polθ. This inhibitor not only elicits the synthetic lethality with *BRCA*-genes previously predicted by genetic studies, but also confers synthetic lethality with defects in the 53BP1/Shieldin DNA repair complex that are a source of PARP inhibitor resistance. As such, this work suggests that Polθ inhibitors not only have clinical potential in targeting *BRCA*-gene defective cancers but could also be used to target PARP inhibitor resistance.

## Results

### Discovery of ART558, a potent and specific small molecule Polθ inhibitor.

We developed a high-throughput DNA primer extension assay based on picogreen incorporation to measure the polymerase activity of full-length (residues 2–2590) Polθ. Using this assay, we screened ~165,000 compounds for their ability to inhibit Polθ, identifying inhibitors with $IC_{50}$ values in the low micromolar range. These latter compounds were further validated and optimized to improve potency and physicochemical properties (to be published elsewhere). From this process we identified ART558 which exhibited a Polθ $IC_{50}$ of 7.9 nM against the isolated full-length enzyme, had good solubility (381 μM (0.16 mg/mL)) and moderate LogD (pH 7.4 = 3.5) (Fig. 1a, b). We found that Polθ activity was entirely dependent on (S)- stereochemistry of the proline ring of ART558, such that the related isomer, ART615, elicited <10% Polθ inhibition at 12 μM, thus serving as a control for further experimentation (Fig. 1b). Mechanistic studies of ART558 revealed non-competitive inhibition with respect to dNTPs and uncompetitive inhibition with respect to DNA, suggesting an allosteric binding site for ART558 within the polymerase catalytic domain of Polθ. (Fig. 1c, d). Through differential scanning fluorimetry, we also found that ART558 elicited thermal stabilisation of Polθ, but only in the presence of DNA (Fig. 1e). In contrast, and as expected, ART615 did not stabilize Polθ to thermal unfolding in either the presence or absence of DNA. ART558 did not inhibit other human DNA polymerases, namely Polα, Polγ, Polη and Polν (Supplementary Table 1). We also tested the compound against a panel of 78 oncology-focussed kinases, as well as PARP1 and PARP2; ART558 showed no significant inhibition of any of these enzymes at 10 μM (Supplementary Materials and Methods). Cellular target engagement of Polθ by ART558 was demonstrated as ART558 increased the residence time of YFP-tagged full-length Polθ at sites of laser-induced DNA damage, whereas ART615 had no effect (Fig. 1f, g). Although this remains to be established by the discovery and assessment of a wider variety of Polθ inhibitors, it is possible that the increased residence time of Polθ on damaged DNA reflects the "trapping" of Polθ on DNA, similar to the concept of PARP1 trapping by clinical PARP inhibitors. We also assessed the effect of ART558 on cellular TMEJ, using PCR and luminescence-based DNA reporter assays that were adapted from a previously described MMEJ assay[13] (Fig. 1h). These assays demonstrated that ART558, but not the control compound ART615, was able to inhibit Polθ-mediated DNA DSB repair with sub micromolar potency, but did not inhibit canonical NHEJ (Fig. 1i, j) further demonstrating excellent selectivity of the molecule. Finally, as genetic inactivation of either the mouse or human homologues of *POLQ* cause radiosensitivity[19–22], we also assessed the specificity of ART558, by assessing whether it was epistatic with *Polq* genetic deletion. In these experiments, we used relatively low doses of ionising radiation (<2 Gy) with minimal effects on survival of wild type cells, so as to maximise the potential for detecting radiosensitivity caused by *Polq* deletion or small molecule inhibition. As expected, deletion of *Polq* caused radiosensitivity in mouse embryonic stem cells (Fig. 1k). ART558 only enhanced radiosensitivity in *Polq* wild-type cells, having no effect in *Polq* null cells (Fig. 1k), suggesting epistasis with *Polq* deletion and an on-target effect. In addition, ART615, the inactive isomer of ART558, did not enhance the radiosensitivity of *Polq* wild type cells (Fig. 1l). Taken together these data suggested that ART558 is a potent and selective inhibitor of the polymerase activity of Polθ, increases retention of Polθ at sites of DNA damage and is active in modulating cellular TMEJ and radiosensitivity.

### BRCA-gene mutations cause sensitivity to ART558.

Genetic inactivation of *POLQ* confers synthetic lethality with *BRCA2* gene defects[17]. We confirmed this genetic synthetic lethal effect using *POLQ* siRNA in isogenic DLD1 cells with or without a truncating mutation in *BRCA2* (DLD1.*BRCA2*^wild-type and DLD1.*BRCA2*^−/− cells[23], previously shown to be PARP inhibitor resistant or sensitive, respectively[24], Fig. 2a). Having validated this system, we used these same cells to demonstrate the ART558 sensitivity of

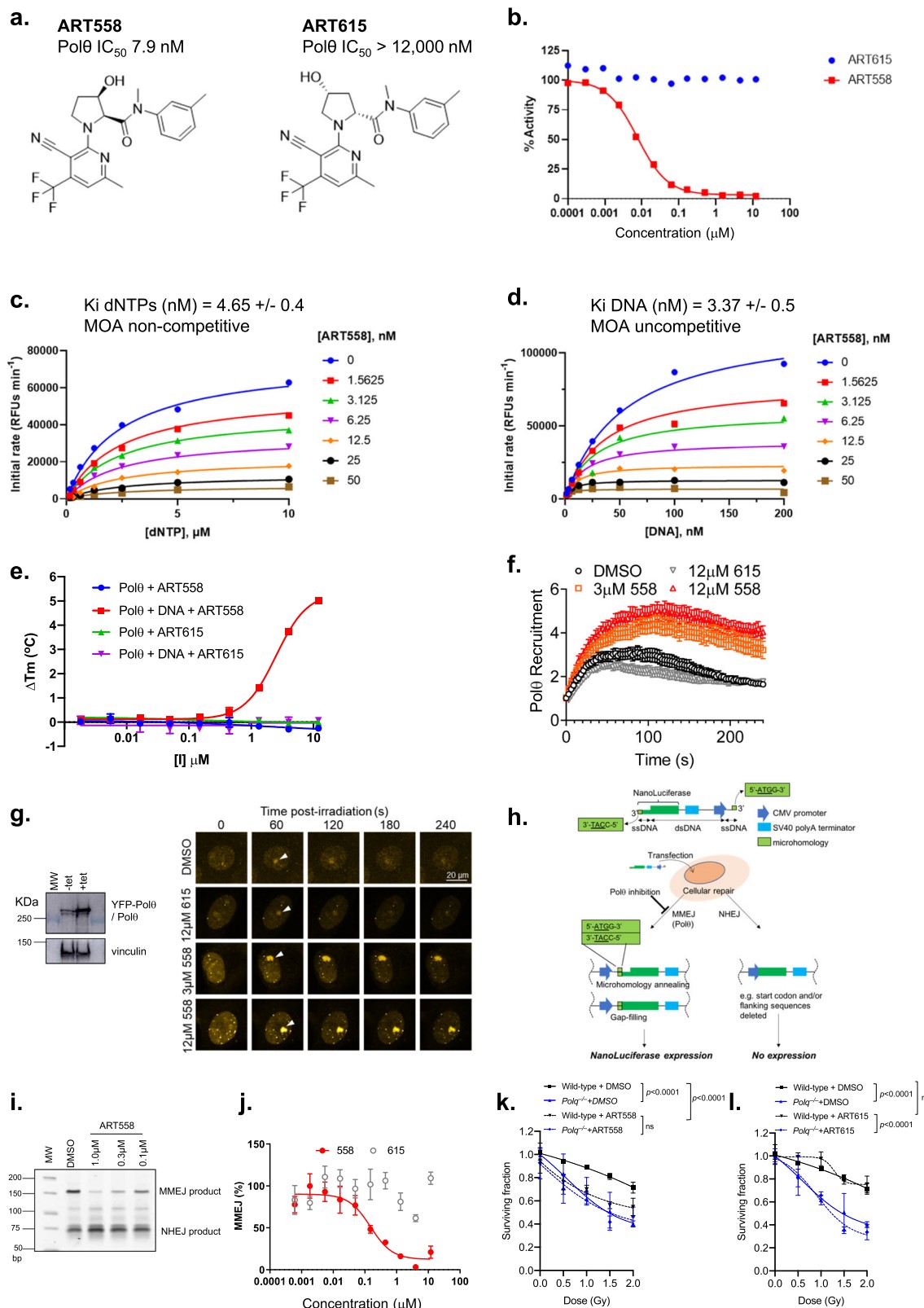

*BRCA2*$^{-/-}$ cells compared to isogenic *BRCA2*$^{wild\text{-}type}$ cells (Fig. 2b). Although loss of *BRCA2* caused sensitivity to ART558, it did not cause profound sensitivity to the structurally related but inactive small molecule ART615 (Supplementary Fig. 1a), although a high concentration (10 μM) of ART615 did have a modest effect in *BRCA2* defective cells, presumably via an off-target mechanism. Genetic inactivation of *POLQ* also causes

sensitivity to PARP inhibitors[17]; this phenotype was replicated with ART558, with a combination of ART558 plus the PARP inhibitor olaparib having a far greater effect on cell survival, culture confluency and apoptosis (as estimated by a caspase 3/7 assay) in DLD1.*BRCA2*$^{-/-}$ cells than in DLD1.*BRCA2*$^{wild\text{-}type}$ cells (Fig. 2c–f). We also found that ART558 exposure elicited a series of DNA damage related biomarkers to a greater degree in

**Fig. 1 ART558 inhibits Polθ polymerase activity. a** Structure and biochemical $IC_{50}$ activity of the DNA polymerase Polθ inhibitor ART558 and its inactive isomer, ART615. **b** Biochemical dose–response curves for ART558 and ART615. **c** dNTP competition assay. ART558 exhibits non-competitive inhibition with respect to dNTPs. **d** DNA competition assay. ART558 exhibits uncompetitive inhibition with respect to DNA. **e** Differential scanning fluorimetry assay. ART558 shows DNA-dependent thermal stabilisation of Polθ, ART615 does not. Each data point represents the mean ± SD from two technical replicates. **f** DNA damage localization assay. YFP-Polθ was recruited to sites of laser-microirradiation after pre-incubation with the indicated compounds or DMSO. Each data point represents the mean ± SEM from n cells (DMSO $n = 28$; 12 μM ART615 $n = 21$; 3 μM ART558 $n = 17$, 12 μM ART588 $n = 27$). ART558 (orange, red), but not ART615 (grey), enhanced YFP-Polθ recruitment indicating direct target engagement. **g** Left: YFP-Polθ expression in U2OS Flp-In Trex cells. Right: images of YFP-Polθ localization to DNA damage. Frames at indicated time points show representative recruitment of YFP-Polθ to damage after compound treatment, scale bar = 20 μm. White arrow indicates micro-irradiation site. **h** Schematic describing a luminescent, extrachromosomal, TMEJ reporter assay. Structure of the extrachromosomal TMEJ repair reporter substrate is shown. **i** ART558 suppresses TMEJ-mediated repair of an extrachromosomal DNA substrate. ART558 suppressed the formation of the MMEJ product, but not the NHEJ product. **j** ART558, but not ART615, inhibits TMEJ-mediated repair of a NanoLuciferease-encoding TMEJ reporter substrate. ART558 inhibited TMEJ in a dose-dependent manner, with a cellular $EC_{50}$ of 150 nM. Data represent mean ± SEM of $n = 4$ (technical replicates) and are representative of biological replicates using indicated compounds. **k, l** Clonogenic survival of *Polq* wildtype and *Polq* null mouse embryonic stem cells after exposure to different doses of ionizing radiation (IR). **k** ART558 elicits radio-sensitivity in *Polq* wildtype but not *Polq* null mouse embryonic stem cells. **l** ART615 does not elicit radio-sensitivity. Cell survival seven days after irradiation is shown. Data are mean ± SD, $n = 3$ Surviving Fractions. Two-way ANOVA with Tukey post hoc test was used to calculate p values, $p = ns$ for p values >0.1.

$BRCA2^{-/-}$ cells compared to isogenic $BRCA2^{wild}$-type cells. These included persistent nuclear γH2AX foci (Fig. 2g), an increase in the total amount of phosphorylated H2AX (as detected by western blotting, Fig. 2h), chromosomal abnormalities as seen in metaphase spreads (Fig. 2i) and increased formation of micronuclei (Fig. 2j). We also noted that ART558 sensitivity was not limited to genetically engineered DLD1.$BRCA2^{-/-}$ cells and was also apparent in tumour cells with a naturally-occurring pathogenic $BRCA2$ mutation, namely CAPAN1 pancreatic ductal adenocarcinoma tumour cells ($BRCA2$ c.6174delT[25], CAPAN1-Parental, Fig. 2k). To confirm the causative relationship between $BRCA2$ status and ART558 sensitivity, we also assessed ART558 sensitivity in a previously described PARP inhibitor resistant (Supplementary Fig. 1b) CAPAN1 daughter tumour cell clone that possesses a reversion mutation that restores the native open-reading frame of $BRCA2$, CAPAN1$^{Revertant}$ [24]. Although CAPAN1$^{Revertant}$ cells still retained some residual ART558 sensitivity, when compared to CAPAN1$^{Parental}$ cells, this was significantly reduced (Fig. 2k). In parallel, we also used a publicly available CRISPR-Cas9 screen dataset[26] to confirm the dependency of CAPAN1 cells upon $POLQ$. Using $POLQ$ CERES scores[27] we found that among 249 tumour cell lines tested, CAPAN1 cells were among the most sensitive, as were a number of BRCA1-mutant tumour cell lines (Supplementary Fig. 1c). Reanalysis of publicly available cancer genomic data also suggested that of the four cancer histotypes that commonly exhibit pathogenic $BRCA1/2$ mutations, $BRCA$-mutant breast cancers tended to express higher POLQ mRNA than for BRCA-gene wild type breast cancers (Supplementary Fig. 1d).

**Combined defects in *BRCA1* and the Shieldin complex that cause PARP inhibitor resistance are associated with Polθ inhibitor sensitivity.** We also observed ART558 synthetic lethality and a combinatorial effect with the PARPi olaparib in previously described[28] isogenic models of $BRCA1$-deficiency (RPE1.$TP53^{-/-}$;$BRCA1^{wild-type}$ vs. RPE1.$TP53^{-/-}$;$BRCA1^{-/-}$ cells, Fig. 3a, b and ID8 $Trp53^{-/-}$, $Trp53^{-/-}$ $Brca1^{-/-}$ or $Trp53^{-/-}$ $Brca2^{-/-}$ cells, Supplementary Fig. 2a, b), as well as ART558 sensitivity in tumour cell lines with endogenous pathogenic $BRCA1$ mutations (COV362[29] ovarian and MDA-MB-436[30] breast tumour cells, Fig. 3c, Supplementary Fig. 2c), suggesting these effects are not specific to $BRCA2$-mutant cells, but also extend to tumour cells with $BRCA1$ mutation (see also Supplementary Fig. 1c). Importantly, at concentrations that elicited cell inhibition in $BRCA1$ or $BRCA2$ mutant cells, ART558 had minimal effects in non-tumour

epithelial cells such as the human mammary epithelial cell lines, MCF10A, MCF12A and HMLE3, or in $BRCA$-gene wild type CAL51 triple-negative breast tumour cells (Fig. 3c). In addition, we also observed ART558 sensitivity in an ex vivo cultured tumour organoid derived from a $BRCA1$-mutant breast cancer, KCL014BCPO, but not in a $BRCA1$ wild type breast cancer organoid cultured under similar conditions (Fig. 3d, e). As expected, the $BRCA1$-mutant breast cancer organoid was sensitive to olaparib (Fig. 3f).

To identify additional determinants of Polθ inhibitor sensitivity, we conducted an ART558/siRNA chemosensitization screen in isogenic RPE1.$BRCA1^{wild type}$ and RPE1.$BRCA1^{-/-}$ cells[28,31–33] (Fig. 3g, Supplementary data 1). In parallel, we also conducted a PARPi sensitivity screen, using olaparib, in the same isogenic cells (Fig. 3h). In each screen, we identified those siRNAs that caused chemosensitization by calculating Drug Effect (DE) Z Scores (see 'Methods') and considered DE Z Scores < −2 (equivalent to two median absolute deviations from the median effect) as representing profound effects worthy of further study. In the ART558 chemosensitization screen in $BRCA1^{wild type}$ cells, siRNA designed to target the HR-associated genes $BRCA1$ and $PALB2$ caused sensitivity (Fig. 3g), along with siRNAs targeting either the Shelterin protein coding gene, $POT1$ or the translesion synthesis-associated gene, $POLH$. Whilst the mechanisms explaining the POT1 or POLH chemosensitization effects are at present not clear, the identification of $BRCA1$ and $PALB2$ was consistent with our data suggesting that ART558 targets HR defective cells. In the parallel olaparib chemosensitization screen in $BRCA1^{wild type}$ cells, we noted that a number of siRNAs that caused olaparib sensitivity in our screen have also been reported as sensitizing hits in a genome-wide CRISPR-Cas9 screen[34]. These include $BRCA1$, $PALB2$, $LIG1$, $ATM$, $SLX4$ and $WEE1$ (Fig. 3h and Supplementary Fig. 2d), the $BRCA1$ and $PALB2$ observations being consistent with their role in HR.

As expected, in RPE1.$BRCA1^{-/-}$ cells, ART558 sensitivity was not enhanced by siRNA designed to target $BRCA1$ or $PALB2$ (most likely because of the pre-existing $BRCA1$ defect) but siRNA designed to target $MAD2L2$ (aka REV7) or $SHLD2$ (FAM35A), both caused profound Polθ inhibitor sensitivity. Both $MAD2L2$ and $SHLD2$ encode components of the MAD2L2/SHLD1/SHLD2/SHLD3 'Shieldin' complex which prevents DNA resection at DSBs[28,31,35–37]. Importantly, loss of Shieldin components cause PARP inhibitor resistance in $BRCA1$ mutant cells[28,33,36,38], suggesting that ART558 might be able to target cells that have become PARP inhibitor resistant via loss of Shieldin. Interestingly, genetic screens in *Polq* null mouse cells have also suggested

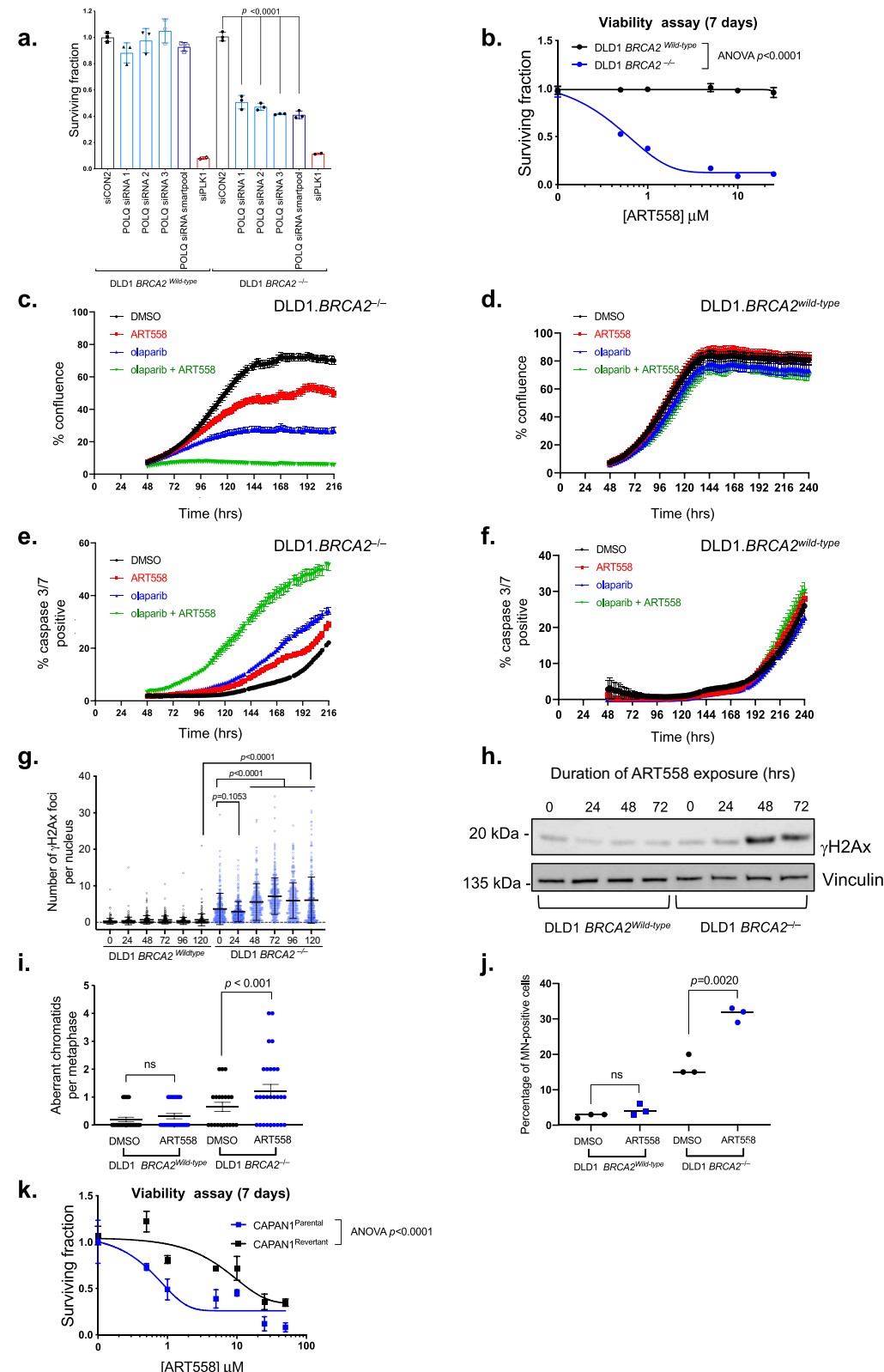

synthetic lethality between elements of this complex and *Polq* genetic deletion[39]. By comparing the ART558 screens in *BRCA1^wild type^* vs. *BRCA1^−/−^* cells, we noted that siRNA targeting Shieldin complex components *SHLD2* and *MAD2L2* sensitised *BRCA1^−/−^* cells to ART558 but had no significant effect in *BRCA1^wild type^* cells, suggesting that combined defects in *BRCA1* and the Shieldin complex might be associated with

ART558 sensitivity. We note that in this screen we used a relatively low concentration of olaparib (a SF_80 concentration eliciting a 20% reduction in cell survival) to maximise the detection of sensitivity-causing effects; this most likely precluded detection of PARPi resistance-causing effects, such as those caused by Shieldin complex defects. The observation that *SHLD2* or *MAD2L2* siRNAs caused sensitivity to ART558 in a *BRCA1*

**Fig. 2 ART558 elicits *BRCA2* synthetic lethality. a** Bar chart illustrating cell survival after siPOLQ transfection in DLD1.*BRCA2*$^{wild\ type}$ or DLD1.*BRCA2*$^{-/-}$ cells. Cells were reverse transfected with siRNAs and after 5 days, cell viability was estimated by CellTiter-Glo. Data are mean ± SD, $n = 3$ Surviving Fractions normalised to siCON2 in each cell line. siRNA targeting *PLK1* was used as a transfection control. Two-tailed Student's *t*-test was used to calculate *p* value. **b** Dose–response survival curves of DLD1.*BRCA2*$^{wild\ type}$ or DLD1.*BRCA2*$^{-/-}$ cells exposed to ART558 for seven days. Cell viability was estimated by CellTiter-Glo. Data are mean surviving fractions ± SD, $n = 3$. Two-way ANOVA with Sídák post hoc test was used to calculate *p* values. **c–f** Incucyte generated confluence (**c**, **d**) or caspase 3/7 reporter fluorescence (**e**, **f**) plots from DLD1.*BRCA2*$^{wildtype}$ or DLD1.*BRCA2*$^{-/-}$ cells exposed to a combination of ART558 and olaparib. Data are mean surviving fraction ± SD, $n = 3$. **g** Dot plot illustrating nuclear γH2AX foci in DLD1.*BRCA2*$^{wild\ type}$ or DLD1.*BRCA2*$^{-/-}$ cells exposed to 5 μM ART558 for the indicated time. Data are means ± SD, $n = 600$. One-way ANOVA with Sídák post hoc test was used to calculate *p* values. **h** Western blot illustrating γH2AX accumulation in DLD1.*BRCA2*$^{wild-type}$ or DLD1.*BRCA2*$^{-/-}$ cells exposed to 5 μM ART558 for the indicated time. Vinculin was used as a loading control. **i** Dot plot illustrating the frequency of aberrant metaphases from DLD1.*BRCA2*$^{wild\ type}$ or DLD1.*BRCA2*$^{-/-}$ cells exposed to DMSO or ART558 for 5 days. $n = 25$ metaphases. Data are means ± SEM *p* value calculated by Wilcoxon rank test. **j** Dot plot illustrating the frequency of micronuclei (MN)-positive cells prepared from DLD1.*BRCA2*$^{wild\ type}$ and DLD1.*BRCA2*$^{-/-}$ cells exposed to DMSO or ART558 for 48 h. $n = 3$ independent experiments. Two-tailed Student's *t*-test was used to calculate *p* value. **k** Dose–response ART558 survival curves in *BRCA2* mutant CAPAN1 parental or revertant cell lines. Cell viability was estimated by CellTiter-Glo reagent after seven days drug exposure. Data are mean surviving fractions ± SD, $n = 3$. Two-way ANOVA with Sídák post hoc test was used to calculate *p* values.

mutant cell line raised the possibility that a Polθ inhibitor could be used to target PARP inhibitor resistance caused by Shieldin complex defects when these occur in *BRCA1* mutant tumour cells. We also noted that *SHLD2* flanks the *PTEN* gene on chromosome 10 and appears to be collaterally lost when *PTEN* is deleted in melanomas and in cancers of the breast or prostate (Supplementary Fig. 3a, b).

To further investigate this possibility in cells with a naturally-occurring pathogenic *BRCA1* mutation, we generated two independent *SHLD2* knockout clones (G1 and D1) from *BRCA1* mutant MDA-MB-436 breast cancer cells (Supplementary Fig. 3c, d). MDA-MB-436 *SHLD2* knockout cells showed ART558 sensitivity and reconstitution of *SHLD2* expression, via SHLD2 cDNA complementation, reversed this effect in both clones (Fig. 3i), confirming a causal relationship. As MDA-MB-436 cells could be established as tumour xenografts in rats, we assessed the ability of a Polθ inhibitor to target established *BRCA1/SHLD2* defective tumours in vivo. ART558 exhibited poor in vitro metabolic stability in rat microsomes (Supplementary Fig. 3f), therefore, we derived a second Polθ inhibitor, ART812, which exhibited good bioavailability and low clearance in rats (Supplementary Fig. 3e, f), as well as eliciting profound Polθ inhibitor sensitivity in MDA-MB-436 *SHLD2* knockout cells (Supplementary Fig. 3g). Dosing of rats bearing established MDA-MB-436 *BRCA1/SHLD2* defective tumours with ART812 resulted in significant tumour inhibition (Fig. 3j, k) and was well-tolerated (Supplementary Fig. 3h). As the activity of the Shieldin complex is controlled by the non-homologous end-joining factor 53BP1 (encoded by *TP53BP1*), and *53BP1, SHLD1* and *SHLD3* defects also cause PARP inhibitor resistance in *BRCA1* mutant cells[38,40,41], we generated PARP inhibitor resistant SUM149 clones with either *SHLD1, SHLD3* or *53BP1* mutations; these each exhibited ART558 sensitivity (Supplementary Fig. 3i) while being resistant to the PARP inhibitor, olaparib (Supplementary Fig. 3j). To further confirm the sensitivity of *BRCA1* and 53BP1 defective lines to Polθ inhibition, we also assessed ART558 sensitivity in previously-validated isogenic mouse embryonic fibroblast cell lines with a hypomorphic *Brca1* allele (*Brca1Δ11*)[38] and *Trp53bp1* defect[42]. We found that whilst the *Brca1* mutation caused a moderate increase in ART558 sensitivity (as might be expected for a hypomorphic *Brca1* allele) and mutation of *Trp53bp1* alone did not cause ART558 sensitivity, the combination of both *Brca1* and *Trp53bp1* mutations was associated with profound ART558 sensitivity (Fig. 3l). We also found that a patient-derived tumour ex vivo culture with low BRCA1 and 53BP1 expression was resistant to olaparib or carboplatin but sensitive to ART558 (Fig. 3m, Supplementary Fig. 3k,l).

We noted that although *BRCA1* mutant models were more sensitive to ART558 than either isogenic *BRCA1* wild type cell lines or non-tumour breast epithelial cells (Fig. 3c), there was clearly a spectrum of responses, with SUM149 cells and *Brca1Δ11* MEFs exhibiting modest sensitivity, when compared to, for example, COV362, MDA-MB-436 or RPE.*BRCA1*$^{-/-}$cells (Fig. 3l and Supplementary Fig. 3i). One explanation for this might be that the *BRCA1* mutations in SUM149 and *Brca1Δ11* MEFs are hypomorphs; in SUM149, the exon 11 truncating mutation (*BRCA1* exon 11 c.2288delT frameshift mutation and loss of the WT *BRCA1* allele) only affects the full length p220 isoform of BRCA1 and not the Δexon11 splice variant, which retains some residual BRCA1 function[30]. Likewise, the *Brca1Δ11* allele imparts resection defects but does not cause a profound Rad51 defect[38]. It, therefore, seems possible that the specific BRCA1 mutation and the hypomorphic nature of the proteins present in *BRCA1* mutant cells could explain their variable sensitivity.

**Polθ inhibitor sensitivity is mediated by DNA nuclease-mediated hyper-resection.** In assessing the mechanistic basis of Polθ inhibitor sensitivity in PARPi-resistant *Brca1*-mutant cells with a 53bp1 defect, we noted that ART558 exposure elicited far greater increases in γH2AX, RPA (Replication Protein A) and pRPA foci in *Brca1Δ11/Trp53bp1*$^{-/-}$ cells compared to *Brca1Δ11* or *Trp53bp1*$^{-/-}$ single mutant cells (Fig. 4a-c). We reasoned that these phenotypes could be caused by the failure to properly process single-stranded (ss)DNA produced by nucleolytic resection of DSB ends[42]. To assess this we measured incorporated BrdU under non-denaturing conditions, where the BrdU signal can only be detected in ssDNA. Nuclear BrdU intensity was increased in *Trp53bp1*$^{-/-}$ and *Brca1Δ11/Trp53bp1*$^{-/-}$ cells to a greater degree than in *Brca1Δ11* cells, with the most significant increase being in *Brca1Δ11/Trp53bp1*$^{-/-}$ cells (Fig. 4d, e). 53BP1 and the Shieldin complex prevent resection in part because of the ability of SHLD2 to bind short 3′ overhangs in DNA via three single-strand binding OB folds[28,31–33]. The preferred substrate of Polθ is also DNA DSBs with relatively short 3′ overhangs which are used as templates to drive TMEJ[13]. Given this, we propose that in *Trp53bp1*$^{-/-}$cells, the absence of 53bp1/Shieldin function leads to an increase in resection and ssDNA when cells are exposed to a Polθ inhibitor. In this case, the resected DNA ends are presumably repaired by a Brca1-mediated process, such that increases in γH2AX and pRPA are relatively modest and cell fitness not profoundly impaired. Conversely when both Brca1 and 53bp1/Shieldin function are impaired, Polθ becomes essential for repairing resected ssDNA caused by the exposure of DSB ends to nucleases due to Shieldin loss. As such, inhibition of Polθ in this

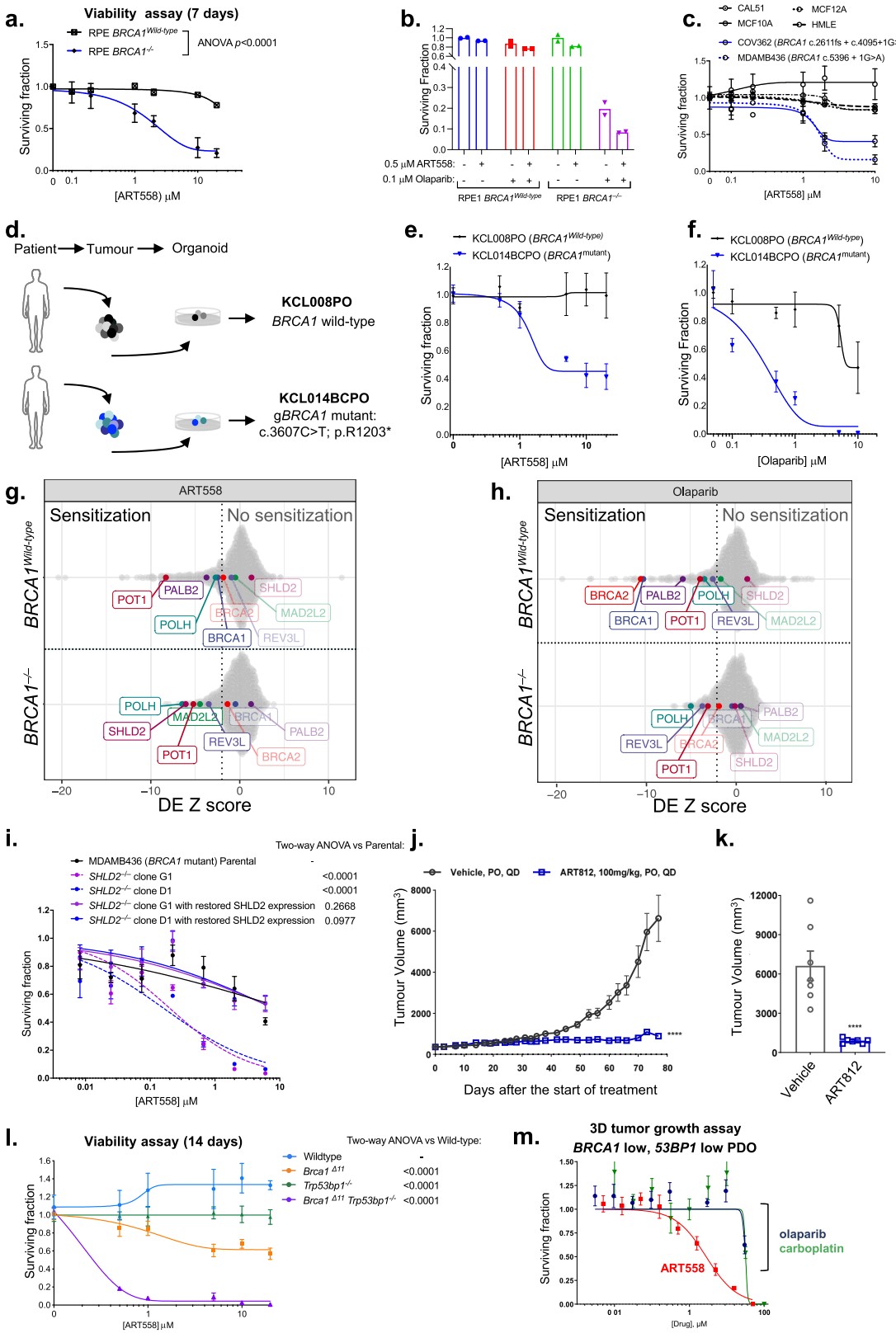

setting leads to a heightened γH2AX and pRPA response and ultimately, synthetic lethality. Cells with combined Brca1 and 53bp1 defects, whilst being PARPi resistant, have only partial but not full restoration of homologous recombination[28,31], providing one possible explanation as to why resected DNA might not be properly processed in this setting. To investigate this, we tested

whether DNA resection at DSBs was enhanced in the presence of a Polθ inhibitor in cells with both *BRCA1* and *53BP1* defects. To do this we used a DSB Inducible via *Asi*SI assay (DIvA assay) where the stable expression of an inducible *Asi*SI-ER fusion restriction enzyme in U20S cells (the *Asi*SI-ER-U20S cell line) generates a resectable DSB at a *Asi*SI site when exposed to

**Fig. 3 Defects in the Shieldin complex cause PARP inhibitor resistance but Polθ inhibitor synthetic lethality. a** Survival curves of RPE1.*BRCA1*$^{wild-type}$ or RPE1.*BRCA1*$^{-/-}$ cells exposed to ART558 for 7 days. Cell viability was estimated by CellTiter-Glo. Data are mean surviving fractions (SF) ± SD, $n = 3$. Two-way ANOVA with Sídák post hoc test was used to calculate $p$ values. **b** Bar plots illustrating SF of RPE1.*BRCA1*$^{wild type}$ or RPE1.*BRCA1*$^{-/-}$ exposed to DMSO, 0.1 μM olaparib, 0.5 μM ART558 or olaparib/ART558 for seven days. Data are $n = 2$. **c** Area under the curve of dose–response ART558 survival curves in *BRCA1* wild type or mutant cell lines. Cell viability was estimated by CellTiter-Glo reagent after 5 days drug exposure. Data are mean SF ± SD, $n = 3$. **d** Overview of breast tumour organoid isolation culture and *BRCA1* mutation. **e** Overview of breast tumour organoid culture. **e, f** Dose–response survival curves of breast tumour organoid cultures exposed to ART558 (**e**) or olaparib (**f**) for 14 days. Cell viability was estimated by CellTiter-Glo. Data are mean SF ± SD, $n = 3$. **g** Drug Effect Z-scores for ART558 or **h** olaparib chemosensitisation screen in RPE1 *BRCA1*$^{-/-}$ or RPE1 *BRCA1*$^{Wild-type}$ cells. **i** Dose–response survival curve for MDA-MB-436 cells (parental), two SHLD2-defective clones (dotted lines) or two SHLD2-defective clones with restored SHLD2 expression. Data are mean SF ± SD, $n = 3$. Two-way ANOVA with Sídák post hoc test was used to calculate $p$ values (**j**). Therapeutic response to ART812 in rats bearing established MDA-MB-436 *BRCA1* mutant, *SHLD2*$^{-/-}$ xenografts. Animals were treated over 76-days with either drug vehicle ($n = 7$) or ART812 (100 mg/kg, $n = 7$). ****$p = $<0.0001 (two-way ANOVA with Sídák multiple comparison test). **k** Tumour volume measurement at the end of study. ****$p = $<0.0001 (two-way ANOVA with Sídák multiple comparison test). **l** Dose–response survival curve for wild-type, *Brca1*$^{Δ11,}$ *Trp53bp1*$^{-/-}$ or *Brca1*$^{Δ11}$*Trp53bp1*$^{-/-}$ MEF cells exposed to ART558 for 14 days. Cell viability was estimated by CellTiter-Glo reagent. Data are mean SF ± SD, $n = 3$. Two-way ANOVA with Sídák post hoc test was used to calculate $p$ values. **m** ART558, olaparib and carboplatin survival curves in a prostate cancer patient-derived organoid culture. PR6512 (see Supplementary Fig. 2i, j). Data are mean SF ± SD, $n = 3$.

4-hydroxytamoxifen (which induces the nuclear localisation of *Asi*SI). In *Asi*SI-ER-U2OS cells, the subsequent resection of the DSB flanking the *Asi*SI site is estimated by the presence of single-stranded DNA at a locus adjacent to the *Asi*SI site, detected by a subsequent *Bsr*G1 restriction digest and real-time PCR reaction[43] (Supplementary Fig. 4a). Using this system, we found that the percentage of ssDNA flanking the *Asi*SI site was increased in cells transfected with both *BRCA1* and *53BP1*-targeting siRNA and exposed to ART558, when compared to similarly treated cells transfected with either control, non-targeting siRNA, or *53BP1* or *BRCA1* siRNA alone (Supplementary Fig. 4b, c). Based on the above data, we reasoned that inhibition of key endonucleases that mediate DNA resection would partially reverse Polθ inhibitor sensitivity in *BRCA1/53BP1* defective cells. Long-range resection, which if excessive can lead to RPA exhaustion and genomic instability, is performed 5′-3′ by either EXO1 or BLM-DNA2[44–48]. EXO1 generates extensive 3′ ssDNA[48,49], whilst the RecQ-helicase BLM unwinds dsDNA[46]. The ssDNA produced is then bound by RPA which protects the 3′ end from DNA2-mediated degradation[46,50,51]. We found that siRNA targeting *Exo1*, *Blm* or *Dna2* partially reversed the sensitivity of ART558 in cells that had both *Brca1* and *Trp53bp1* mutations (Fig. 4f, Supplementary Fig. 4d, e). This observation was further validated by using individual siRNAs (Fig. 4g, Supplementary Fig. 4f) or CRISPR Cas9 mutagenesis (Supplementary Fig. 4g) targeting *Blm* or *Exo1*. Furthermore, we found that siRNA targeting *Exo1*, *Blm* or *Dna2* reduced the ART558-mediated induction of γH2AX foci (Fig. 4h). The same was true for the accumulation of pRPA after ART558 exposure (Fig. 4i). Taken together, these phenotypes were consistent with resection via *Exo1* or *Blm-Dna2* being a cause, at least in part, of the ART558 sensitivity phenotype.

## Discussion

The observations described above demonstrate the discovery of a potent and selective small molecule Polθ inhibitor that not only elicits *BRCA*-gene synthetic lethality, but also targets cells with PARPi resistance caused by 53BP1/Shieldin defects. As far as we are aware, the discovery of ART558 provides the first evidence that a DNA repair related DNA polymerase can be targeted with a specific small molecule and the synthetic lethality with *BRCA1* or *BRCA2* provides a rare example of where a synthetic lethality identified via genetic means can be reproduced with a small molecule inhibitor. The ART558 sensitivity in *BRCA2* mutant cells confirms the genetic synthetic lethality seen in previous studies[17,52]. Previously, others have suggested that the *BRCA2/POLQ* synthetic lethality is caused by the loss of TMEJ and the subsequent chromosomal abnormalities that emerge when both

HR and TMEJ are impaired. The mitotic defects and micronuclei that we observed in *BRCA2* mutant cells upon ART558 exposure are consistent with this mechanistic hypothesis[17]. By identifying the synthetic lethality between ART558 and 53BP1/Shieldin defects in *BRCA1* mutant cells, our data suggest that Polθ inhibition might also provide a route to targeting some forms of PARP inhibitor resistance (Fig. 4j). For example, using a Polθ inhibitor in combination with or after a PARP inhibitor in patients with *BRCA1* mutant cancers might prevent the emergence of otherwise drug resistant 53BP1/Shieldin defective tumour cell clones. Furthermore, Polθ inhibitors might have added utility when used in concert with PARP inhibitors or platinum salts, given: (i) the proposed role of microhomology-mediated DNA repair processes in forming *BRCA*-gene reversion mutations that cause therapeutic resistance[53]; and (ii) the central role of Polθ in TMEJ, a form of microhomology-mediated DNA repair[8]. Although a causative link between TMEJ, Polθ and *BRCA*-gene reversion mutations still remains to be confirmed, this suggests that the potential for Polθ inhibitors might extend beyond *BRCA1* mutant cancers with 53BP1/Shieldin complex defects.

We also note other questions that if addressed might inform how Polθ inhibitors are best used clinically. After the original identification of BRCA/PARP inhibitor synthetic lethality in 2005[54,55], subsequent work determined that: (i) some PARP inhibitor resistance mechanisms are shared between *BRCA1* and *BRCA2* defective tumour cells (e.g. reversion mutations in either *BRCA1* or *BRCA2*, upregulation of PgP); (ii) other resistance mechanisms are specific for cells with *BRCA1* mutations (e.g. loss of 53BP1/Shieldin); and (iii) clinically, the penetrance of the BRCA-gene/PARP inhibitor synthetic lethal effect is incomplete[56] with some patients, especially those with advanced disease, showing either de novo or acquired PARP inhibitor resistance, despite the presence of a pathogenic *BRCA*-gene mutation[53]. It seems reasonable to think a similar scenario might also be the case for Polθ inhibitors, where some mechanisms of resistance might be more relevant in a *BRCA1*-mutant vs. a *BRCA2*-mutant setting or where different *BRCA*-gene pathogenic mutations could have differing effects on Polθ inhibitor sensitivity. Understanding whether this is the case or not could inform patient stratification approaches and the identification of predictive biomarkers to be used alongside Polθ inhibitors. We also note that understanding how Polθ inhibitors could be used within drug combination regimens might be critical in informing their clinical development. In general, the clinical delivery of PARP inhibitor/DNA-damaging chemotherapy combinations has proven challenging due to the common manifestation of

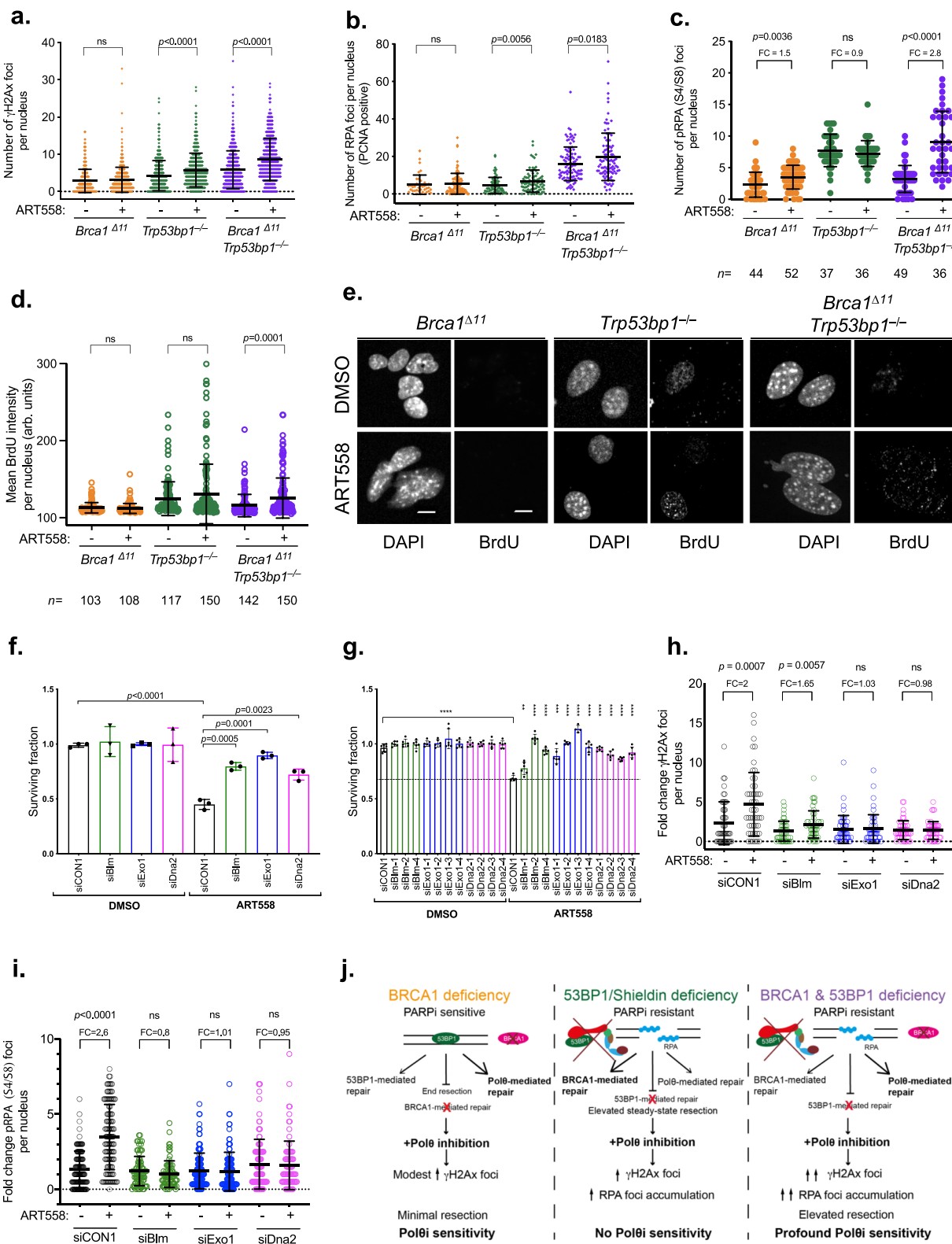

dose-limiting toxicities that are less frequent when single agent PARP inhibitors are used[57]. Although this remains to be established, it seems possible that Polθ inhibitors might be somewhat more suitable for combination with DNA-damaging chemotherapies in a way PARP inhibitors are not.

## Methods

**Synthesis of ART558, ART615 and ART812.** Commercially available reagents were used without purification and reactions were carried out under uncontrolled atmosphere unless otherwise stated. The typical mass spectrometer used for mass-directed HPLC was a Waters 3100 which detected masses between 100 and 700 g/mol. Full synthesis details are provided in Supplementary Methods.

**Fig. 4 DNA damage-related phenotypes in *Brca1*$^{\Delta 11}$;*Trp53bp1*$^{-/-}$ cells exposed to ART558. a** γH2AX foci per nucleus in *Brca1*$^{\Delta 11}$, *Trp53bp1*$^{-/-}$ or *Brca1*$^{\Delta 11}$; *Trp53bp1*$^{-/-}$ MEFs following ART558 exposure. Data are means ± SD, $n = 500$. **b** RPA foci per PCNA-positive nucleus in *Brca1*$^{\Delta 11}$, *Trp53bp1*$^{-/-}$ or *Brca1*$^{\Delta 11}$; *Trp53bp1*$^{-/-}$ MEFs following ART558 exposure. Data are means ± SD, $n = 300$. **c** pRPA (S4/S8) foci per nucleus in *Brca1*$^{\Delta 11}$ or *Brca1*$^{\Delta 11}$;*Trp53bp1*$^{-/-}$ MEFs following ART558 exposure. One-way ANOVA with Tukey post hoc test was used to calculate $p$ values (**a–c**). **d** Nuclear intensity of BrdU in ssDNA in *Brca1*$^{\Delta 11}$, *Trp53bp1*$^{-/-}$ or *Brca1*$^{\Delta 11}$;*Trp53bp1*$^{-/-}$ MEFs following ART558 exposure. Data represents percentage of cells with BrdU foci, $n = 3$ independent experiments. Two-tailed Student's $t$ test was used to calculate $p$ values. **e** Representative images of BrdU foci induction following ART558 exposure in *Brca1*$^{\Delta 11}$, *Trp53bp1*$^{-/-}$ or *Brca1*$^{\Delta 11}$;*Trp53bp1*$^{-/-}$ MEFs, scale bar = 10 µm. **f** Bar plots illustrating SF of *Brca1*$^{\Delta 11}$;*Trp53bp1*$^{-/-}$ MEFs transfected with the indicated siRNA SMARTpools and exposure to DMSO or 5 µM ART558 for 4 days. SF ± SD are shown relative to median DMSO siCON1, $n = 3$. Two-tailed Student's $t$ test was used to calculate $p$ values. **g** Bar plots illustrating surviving fraction of *Brca1*$^{\Delta 11}$;*Trp53bp1*$^{-/-}$ MEFs transfected with the indicated individual siRNA and exposure to DMSO or 5 µM ART558 for four days. Surviving fractions ± SD are shown relative to median DMSO siCON1, $n = 3$. Two-tailed Student's $t$ test was used to calculate $p$ values, ** = 0.0021, *** = 0.0002, ****<0.0001. **h** Fold change in γH2AX foci per nucleus in *Brca1*$^{\Delta 11}$;*Trp53bp1*$^{-/-}$ MEFs following ART558 exposure after transfection with the indicated siRNAs. Data are means ± SD, $n = 50$ nuclei per condition. Two-tailed Student's $t$ test was used to calculate $p$ values. **i** Fold change in pRPA (S4/S8) foci per nucleus in *Brca1*$^{\Delta 11}$;*Trp53bp1*$^{-/-}$ MEFs following ART558 exposure after transfection with the indicated siRNAs. $n = 3$. Two-tailed Student's $t$ test was used to calculate $p$ values. **j** A model for the proposed mechanism driving the sensitivity of BRCA1 and 53BP1 deficient cancers to Polθi.

**DNA Complex preparation.** All oligonucleotides (Supplementary Methods) were purchased from Integrated DNA Technologies and made up to a concentration of 100 µM in annealing buffer (20 mM Tris-HCl pH 7.5, 50 mM NaCl). Polθ substrate was generated using a final concentration of 22 µM of oligonucleotide 1 and 20 µM oligonucleotide 2 in annealing buffer. A slight excess of Oligonucleotide 1 was used to ensure a final annealed substrate concentration of 20 µM. Polθ product was generated using a final concentration of 20 µM oligonucleotides 2 and 3 in annealing buffer. For both substrate and product generation, 50 µL aliquots were heated to 95 °C in a heating block for 5 min before the heating block switched off and the reaction left to cool to room temperature. T8/P6 DNA was generated by mixing equal volumes of the T8 and P6 stocks to give a concentration of 1 mM annealed DNA. Fifty µL aliquots were heated to 85 °C in a heating block for 5 mins before the heating block was switched off and the reaction left to cool to room temperature.

**Biochemical IC$_{50}$ assays.** Polα was purchased from CHIMERx, Polγ from Abcam, and Polη from Enzymax. Polη (aa192–863) was custom synthesized by PeakProteins following the literature precedent from[58]. Picogreen was purchased from ThermoFisher and Deoxynucleotide solution mix (dNTP) from New England Biolabs. Recombinant human full length Polθ (fl-Polθ) was produced using the Bac-to-Bac Baculovirus expression system (Thermo Fisher Scientific) using protocols modified from[8,59]. Full synthesis details are provided in Supplementary Methods.

Compounds were dispensed into black 384-well Proxiplates (PerkinElmer) using and Echo 550 (Labcyte). For each compound, 60 nL of a half-log dilution series was dispensed starting from either a 12 or 1.2 mM top concentration of compound in DMSO. This gave final in-assay top concentrations of 120 or 12 µM, respectively and 1% (v/v) DMSO concentration.

Reactions were performed at room temperature and with the exception of Polη, in freshly prepared assay buffer (25 mM Tris-HCl pH 7.5, 12.5 mM NaCl, 0.5 mM MgCl$_2$, 5% (v/v) glycerol, 0.01% v/v Triton x-100, 0.01% (w/v) Bovine γ-Globulin, 1 mM dithiothreitol). Polη reactions were carried out in modified assay buffer (25 mM Tris-HCl pH 8.0, 50 mM NaCl, 1.5 mM MgCl$_2$, 5% (v/v) glycerol, 0.01% (v/v) Triton X-100, 0.01% (w/v) Bovine γ-Globulin, 1 mM dithiothreitol). 3 µL of 2× polymerase (4 nM fl-Polθ, 4 ng/µL Polα, 4 nM Polγ, 3 nM Polη or 2 nM Polη in assay buffer) and 3 µL of 2× substrate mix (100 nM DNA and 40 µM dNTPs in assay buffer) were added separately to plates that had been pre-dispensed with compound using a Tempest liquid handler (Formulatrix). The plates were covered and left to incubate for 1 h at room temperature, before the enzymatic reaction was quenched by the addition of 4 µL of stop buffer (1:80 dilution of PicoGreen in 25 mM Tris-HCl pH 7.5, 10 mM EDTA) using a Tempest liquid handler (Formulatrix). After the addition of stop buffer, the plates were covered and incubated for 90 min before being read on a PHERAstar FS plate reader (BMG Labtech) using an FI optic module containing a 485 nm excitation filter and a 520 nm emission filter. The gain was set to 50, focal height 11.2 mm, positioning delay of 0.1 s and 20 flashes per well or a CLARIOstar Plus (BMG Labtech) using the default optical settings for fluorescein and the auto gain/focus settings.

IC$_{50}$ data were analysed in Abase (IDBS) or Genedata Screener (Genedata).

**DNA intercalation assay.** To test whether the Polθ inhibitors were causing signal interference by intercalating DNA, ART558 was dispensed into 384-well black low volume non-binding plates (Greiner). For each compound, 100 nL of a 1.5-fold dilution series was dispensed starting from a 12 mM top concentration of compound in DMSO using the D300e digital dispenser (Tecan). A DMSO control was also included. Wells were normalised and backfilled with DMSO such that all wells contained 1% (v/v) after the addition of 10 µL of Polθ DNA product solution. 10 µL of a solution containing 200 nM Polθ product DNA in assay buffer (25 mM Tris-HCl pH 7.5, 12.5 mM NaCl, 0.5 mM MgCl$_2$, 5% (v/v) glycerol, 0.01% (v/v) Triton x-100, 0.01% (w/v) Bovine γ-Globulin, 1 mM dithiothreitol) and 5 µL of detection reagent (25 mM Tris-HCl pH 7.5, 10 mM EDTA, 2.5% (v/v) PicoGreen) were dispensed separately using a Tempest liquid handler (Formulatrix). Plates were then read on the CLARIOstar Plus (BMG Labtech) using the settings described above. Two known DNA intercalators: mitoxantrone and doxorubicin were used as positive controls. IC$_{50}$ data were analysed using GraphPad Prism V.8.4.2 (GraphPad Software Inc, San Diego, CA).

**Mechanism of action assays.** ART558 was dispensed at 6 different concentrations (50, 25, 12.5, 6.25, 3.125 and 1.563 nM) along with a DMSO control into 384-well black low volume non-binding plates (Greiner) as described above using the D300e digital dispenser (Tecan). Wells were normalised and backfilled with DMSO such that all wells contained 1% (v/v) DMSO in the final assay volume. 5 µL/well of 8 concentrations of Polθ substrate DNA (200, 100, 50, 25, 12.5, 6.25, 3.125 and 1.563 nM) or dNTP (10, 5, 2.5, 1.25, 0.625, 0.3125, 0.15625 and 0.078125 µM) were manually dispensed into the appropriate wells using a VIAFLO automated pipette (Integra). The plates were then loaded onto a Tempest liquid handler (Formulatrix) and 5 µL of 4.8 nM pd- Polθ and the second substrate needed for the reaction (either 20 µM of dNTPs or 200 nM of DNA Polθ substrate) was dispensed into all wells. To stop the reaction, 5 µL of a solution containing 25 mM Tris-HCl pH 7.5 and 20 mM EDTA was added at 6 timepoints (0, 5, 15, 30, 45 and 60 min) using the Tempest's time delay function. The plates were covered during the time course to prevent evaporation. After completing the assay, 5 µL of detection reagent (25 mM Tris-HCl pH 7.5 and 2.5% (v/v) PicoGreen) was dispensed into the wells using a Tempest liquid handler (Formulatrix) and plates subsequently read on the CLARIOstar Plus (BMG Labtech) using the settings described above. For details of model fitting, see Supplementary Methods.

**Differential scanning fluorimetry.** For each compound (ART558 and ART615) 100 nL of a half-log dilution series was dispensed starting from a 1.2 mM top concentration of compound in DMSO into a 384 Well Skirted PCR Plate (FrameStar) as described above using the D300e digital dispenser (Tecan). This gave a final in-assay top concentration of 12 µM. Wells were normalised and backfilled with DMSO such that all wells contained 1% (v/v) DMSO in the final assay volume. All experiments were carried out in freshly prepared DSF buffer (25 mM HEPES pH 7.5, 12.5 mM ammonium acetate, 0.5 mM MgCl$_2$, 5% (v/v) glycerol, 5% (v/v) DMSO, 1 mM dithiothreitol). 5 µL of 2× polymerase Polq (4 µM), 2 µL of 5× T8/P6 DNA (25 µM) and 3 µL of 3.33× protein thermal shift dye (ThermoFisher) were added to the appropriate wells of the plate using a Tempest liquid handler (Formulatrix). Wells were backfilled with buffer to ensure that each reaction contained a final volume of 10 µL. The plate was sealed with MicroAmp™ Optical Adhesive Film (ThermoFisher) and centrifuged at $900 \times g$ for 1 min. The plates were then heated & fluorescence measured using a Viia7 qPCR machine (ThermoFisher) and the following thermal profile:

Step 1, Temp: 20 °C, Time: 2 min
Step 2, Temp: 99 °C, Time: 2 min
Ramp mode: Continuous
Ramp rate: Step 1: 1.9 °C/s, Step 2: 0.05 °C/s
Optical Filters—Excitation Filter: ×4 (580 ± 10); Emission Filter: m4 (623 ± 14)

In each experiment, the Tm of Polq in the presence of compound alone or compound + T8/P6 DNA was determined in quadruplicate, along with control wells containing Polq, T8/P6 DNA or Polq + T8/P6 DNA. All Tms were calculated as the inflection point of the melting curve using the derivative analysis function of the ViiA™ 7's PTS software.

**Recruitment of Polθ to laser localized DNA Damage.** Generation of YFP-Polθ FlpIN Trex U2OS cells: Polθ was amplified by PCR using primers F: 5′-GGG GAC AAG TTT GTA CAA AAA AGC AGG CTT CAT GAA TCT TCT GCG TCG

GAG TGG and R: 5′-GGG GAC CAC TTT GTA CAA GAA AGC TGG GTA TTA CAC ATC AAA GTC CTT TAG CT from Addgene plasmid #73132. The PCR product was cloned into pDON221 using a BP clonase II reaction as per the manufacturers instruction to make Polθ-pDON221. Polθ-pDON221 was cloned into YFP-pcDNA5TO/FRT-DEST using an LR clonase II reaction as per manufacturer's instructions to make YFP- Polθ -pcDNA5TO/FRT.

Inducible YFP-Polθ cell lines were generated using the Flp-In T-REx system (Invitrogen) as per manufacturer's instruction. YFP- Polθ -pcDNA5TO/FRT was co-transfected with the pOG44 vector (Flp recombinase) into U2OS host cell lines (kind gift of Daniel Durocher, LTRI Toronto). The host cell line was cultured in DMEM supplemented with 15.5 μg/ml zeocin (Invitrogen) and 5 μg/ml blasticidin (Invitrogen). Polθ integration was selected with 250 μg/ml hygromycin B (ThermoScientific).

Inducible YFP-Polθ FlpIN Trex U2OS cells were cultured in DMEM media (PAN-Biotech) supplemented with 10% Tet-free FBS (Takara Clontech), 1% PenStrep (Sigma), 8 μg/ml blasticidin (Life Technologies) and 100 μg/ml Hygromycin B (Life Technologies) under normal growth conditions (37 °C, 5% $CO_2$). Expression of the transgene was induced by 1 μg/ml tetracycline (Fisher Scientific).

YFP-Polθ FlpIN Trex U2OS cells were grown for 48 h in tetracycline to induce YFP-Polθ expression. Cells were split into fresh antibiotic-free media (DMEM/ 10% Tet-free FBS, 1 μg/mL tetracycline) supplemented with 10 μM 5-bromo-2′-deoxyuridine (Sigma), for pre-sensitisation, and seeded to individual wells of an 8-well Lab-Tek chamber slide (VWR) for a further 24 h. Cells were supplemented with compound at indicated concentrations, or DMSO, for 2 h. Chambers were mounted on the stage of an Andor Revolution Spinning Disk Confocal Microscope and cells were visualized with a Nikon CFI Plan apo 60 ×1.45 NA objective using Andor iQ3 software for image capturing. Throughout the experiment, cells were maintained at 5% $CO_2$, and 37 °C using a live cell environmental chamber (OKOlab). Damage was induced at a preselected region of interest (ROI) 5 px × 5 px (1.8 μm × 1.8 μm), in an individual nucleus using the 405 nm laser (100% power, 50μs dwell time, 100 repeats) and YFP-Polθ imaged using the 515 nm laser (6% power, 300 ms exposure, 300 gain, 2 frame averaging). Images were taken before targeting and at 3 s intervals after targeting to detect the Polθ recruitment to the ROI over 240 s. ImageJ was used to quantify the intensity of the YFP signal at the ROI and at an equivalent area in a non-targeted region of the nucleus (background) at each timepoint to account for different expression levels between cells and photobleaching over time. Graphs show [YFP Intensity at the ROI / background YFP intensity in the same nucleus] as a function of time in arbitrary units for the indicated number of cells in each condition.

**Ionizing radiation (IR) sensitivity assay.** Wild-type and *Polq* knockout mouse embryonic stem cells[22] were seeded at low-density on p60 plates and treated with 6 μM ART558, 6 μM ART615 or DMSO three hours prior to exposure to IR, using an YXlon X-ray generator (YXlon International). Cells were left to grow for seven days, after which plates were fixed with 0.9% NaCl and stained with methylene blue. Surviving colonies were counted, the survival of irradiated cells was calculated relative to the cloning efficiency of non-exposed cell-lines (0 Gy), which was set to 1.0 for each individual sample.

**PCR-based TMEJ repair assay.** HEK293 cells (ATCC) were cultured in MEM Eagle media (PAN-Biotech) supplemented with 10% (v/v) foetal bovine serum (FBS) (PAN-Biotech) under normal growth conditions (37 °C, 5% $CO_2$).

The repair substrate was generated essentially as described in ref. [16] and comprises a 597 bp duplex DNA with flanking 45 nucleotide 3′ overhangs each with 4 nucleotide terminal microhomologies on complementary DNA strands. The assay was performed essentially as described in ref. [16]. Briefly, 200 × 10³ HEK293 cells were incubated at 37 °C for 1 h with compound at indicated concentrations, or DMSO. Cells were resuspended in supplemented SF nucleofection solution (Lonza) containing the TMEJ repair substrate and carrier plasmid pmaxGFP (Lonza) at a ratio of 20 μl SF: 150 ng of repair substrate: 1 μg pmaxGFP: 200 × 10³ cells. DNA was introduced by nucleofection using the program CM-130 on the 4D nucleofector X unit (Lonza) in a 16 chamber nucleocuvette. Two chambers were used per condition, and subsequently combined. Cells were recovered into media containing compound or DMSO and incubated at 37 °C for 1 h. Cells were washed once in PBS and then incubated for 15 min at 37 °C in HBSS (Gibco) with 12.5 U benzonase. Cells were washed twice in PBS and resuspended in 200 μL PBS. Genomic DNA was isolated using the QIAamp DNA Mini Kit (Qiagen) as per the manufacturer's instructions. The polymerase chain reaction (PCR) was carried out using the KOD Hot Start Polymerase kit (Merck), primers S1F 5′-CTT ACG TTT GAT TTC CCT GAC TAT ACAG and S2R 5′-AGC AGG GTA GCC AGT CTG AGA TGGG (Sigma) and 200 ng gDNA template per reaction in a Mastercycler Nexus thermocycler (Eppendorf) with the following programme: 95 °C for 2 min, 40 cycles of [95 °C for 20 s, 64 °C for 10 s, 70 °C for 10 s], 70 °C for 1 min. PCRs were also run on synthesised plasmid templates to generate the expected MMEJ and NHEJ repair products, confirming band specificity. Samples were resolved on a 6% polyacrylamide TBE gel (Life Technologies). Gels were stained with SYBRSafe DNA gel stain (Invitrogen) and imaged on an Amersham Imager 600RGB.

**NanoLuciferase TMEJ repair assay.** The NanoLuciferase TMEJ repair reporter substrate has been engineered to express a functional NanoLuciferase reporter protein only when TMEJ has been correctly performed. The linear DNA substrate comprises an inverted expression locus in which, the gene encoding Nanoluciferase is located upstream of a CMV promoter and is not expressed. Adapting the structure of the TMEJ substrate described in ref. [16] the molecule has a dsDNA core flanked by 45-nt ssDNA 3′-overhangs, each with 4-nt terminal microhomology which encompass the initiator methionine of Nanoluciferase. The locus was gene synthesised (GeneArt) placing the NanoLuciferase gene upstream of a CMV promoter. Silent mutations introduced a *XhoI* restriction site in the NanoLuciferase gene and ensured a single *Hind*III site in the promoter. Oligonucleotides were annealed to generate ssDNA/dsDNA caps with a 45-nt ssDNA overhang containing 4nt terminal microhomology (5′-ATGG (right)/5′-CCAT (left)) and a *XhoI* (left) or *Hind*III (right) complementary overhang. The dsDNA core of the reporter was excised by *XhoI* and *Hind*III and ligated to the caps to generate a single linear DNA molecule which constitutes the transfectable TMEJ repair substrate.

The use of intermolecular or intramolecular TMEJ repair, via the terminal 4-nt of microhomology, places the gene downstream of the promoter and restores the initiator ATG codon (embedded in the microhomology) necessary for expression of the full-length reporter protein. Non-MMEJ repair which compromises the presence or integrity of the ATG codon (e.g. NHEJ) will not result in NanoLuciferase expression. A schematic of the substrate and assay concept is outlined in Fig. 1H.

To assess titratable inhibition of TMEJ, compounds were dispensed into a white 384-well microplate (Costar) using the D300e digital dispenser (Tecan) to generate a 10-point dose–response curve (top concentration 12 μM, dilution factor 3), with a backfilling step included to equalise the final DMSO concentration to 0.1% (v/v).

HEK293 cells were trypsinised, washed, and resuspended in fresh media. After counting, cells were resuspended in supplemented SF nucleofection solution (Lonza) containing the NanoLuciferase TMEJ repair substrate and FireFly luciferase plasmid (Promega) at a ratio of 20 μL SF: 2 μg NanoLuciferase substrate: 800 ng FireFly plasmid: 200 × 10³ cells. DNA was introduced by nucleofection using the program CM-130 on the 4D nucleofector X unit (Lonza) in a single cuvette. Cells were recovered into fresh media. 4 × 10³ cells (25 μL of cell suspension) were seeded per well using a MultiFloFX (BioTek), directly onto compound pre-plated in the 384-well-plate and incubated for 24 h at 37 °C. Firefly and NanoLuciferase levels were detected using the Nano-Glo Dual-Luciferase Reporter Assay system (Promega) as per the manufacturer's instructions, and luminescence was measured with a Clariostar plate reader (BMG Labtech), using the manufacturer's protocols 'FireFly' and 'NanoLuciferase'. In each experimental well the NanoLuciferase signal was normalised to the Firefly signal, which served as a measure of both cell density and transfection efficiency and then normalised to the DMSO control. Promega was the source of the NanoLuc® technology and the Modified NanoLuc® Polynucleotides. The modified NanoLuc® polynucleotide sequence used in this paper, which encodes the NanoLuc® reporter protein upon TMEJ-mediated repair, differs from the canonical sequence by two nucleotide substitutions (T72C, A75G). These substitutions are functionally silent. Artios Pharma was authorised by Promega to generate the modified NanoLuc® polynucleotide.

**Cell lines.** SUM149 cells (original source: Asterand, described in ref. [24], were cultured under normal growth conditions (37 °C, 5% $CO_2$) and passaged at 80% confluency. Growth medium consisted of Ham's F-12 medium (Gibco) supplemented with 5% (v/v) heat inactivated foetal bovine serum (FBS) (Sigma-Aldrich), 10 μg/mL insulin (Sigma-Aldrich), 0.5 μg/mL hydrocortisone (Sigma-Aldrich) and penicillin/streptomycin (Gibco). CAL51 cells (source: DSMZ), COV362 (source: ECACC) and MDA-MB-436 (source: ATCC) cells were maintained in high glucose- and GlutaMAX-supplemented DMEM (Gibco, Thermo Fisher Scientific) + 1% (v/v) penicillin–streptomycin (Thermo Fisher Scientific) and 10% (v/v) heat inactivated fetal calf serum (Gibco) at 37 °C, 5% $CO_2$. RPE1 $TP53^{-/-}$ $BRCA1^{Wild-Type}$, RPE1 $TP53^{-/-}$ $BRCA1^{-/-}$,cells were generated as previously described[28] (gift from D. Durocher). Cells were cultured in high glucose- and GlutaMAX-supplemented DMEM (Gibco, Thermo Fisher Scientific) + 1% (v/v) penicillin–streptomycin (Thermo Fisher Scientific) and 10% (v/v) heat inactivated fetal calf serum (Gibco) at 37 °C, 5% $CO_2$. DLD1 $BRCA2^{Wild-type}$ and DLD1 $BRCA2^{-/-}$ cells (source: Horizon Discovery Inc.) were cultured in high glucose- and GlutaMAX-supplemented DMEM (Gibco, Thermo Fisher Scientific) + 1% (v/v) penicillin–streptomycin (Thermo Fisher Scientific) and 10% (v/v) heat inactivated fetal calf serum (Gibco) at 37 °C, 5% $CO_2$. CAPAN1$^{Parental}$ (source: ATCC) and CAPAN1$^{Revertant}$ (described and characterised in ref. [24]) cells were maintained in high glucose- and GlutaMAX-supplemented DMEM (Gibco, Thermo Fisher Scientific) + 1% (v/v) penicillin–streptomycin (Thermo Fisher Scientific) and 20% (v/v) heat inactivated fetal calf serum (Gibco) at 37 °C, 5% $CO_2$. Mouse embryonic fibroblasts (MEFs) were generated as previously described[42]. Cells were cultured in high glucose- and GlutaMAX-supplemented DMEM (Gibco, Thermo Fisher Scientific) + 1% (v/v) penicillin–streptomycin (Thermo Fisher Scientific) and 15% (v/v) heat inactivated fetal calf serum (Gibco) at 37 °C, 5% $CO_2$. ID8 cells were generated as previously described[60]. Cells were cultured in high glucose- and GlutaMAX-supplemented DMEM (Gibco, Thermo Fisher Scientific) + 1% penicillin–streptomycin (Thermo Fisher Scientific), 1× AOF ITS supplement (Sigma-Aldrich) and 4% (v/v) heat

inactivated fetal calf serum (Gibco) at 37 °C, 5% $CO_2$. Cells were STR typed to confirm identity and verified to be mycoplasma-free prior to the study.

SHLD2 KO clones in MDA-MB-436 cells, were generated by Oxford Genetics. Briefly, synthetic guide RNAs (sgRNA) for CRISPR/Cas9 were designed to specifically target a key coding exon of the gene of interest. Pools of cells carrying the edited gene were generated by transient co-transfection of the sgRNA complexed with CRISPR/Cas9 protein. Single cells were isolated, and the targeted exon was sequenced by Sanger sequencing. Selected clones with out-of-frame insertion/deletions in all alleles were expanded and validated by PCR followed by high-throughput sequencing. Two SHLD2 KO MDA-MB-436 clones (D1 and G1) were generated. The restoration of SHLD2 protein expression in SHLD2 KO MDA-MB-436 cell line was achieved by lentiviral vector transduction. Briefly, for lentiviral production, $8 \times 10^6$ HEK-293T cells were seeded into a 10 cm plate (10 ml cell media, containing heat inactivated (HI) FBS and 1% Pen-strep (Gibco)) and, 24 h later, transfected with the plasmid mix. For SHDL2 expressing lentiviral vector and GFP controls we used Ex-A8388-Lv105 and Ex-EGFP-Lv105 (Labomics), respectively. For the transfection, 20 µg of LV packaging plasmid mix (ABM) and 20 µg transfer plasmid Ex-A8388-Lv105 or Ex-EGFP-Lv105 were combined with lipofectamine 3000 (Thermofisher) in Optimem (Gibco) at a 1:3 DNA: lipofectamine ratio and used as per the manufacturer's instructions. After 16 h, the media was exchanged. 46 h after the transfection, cell media containing lentiviral particles were collected from the plates: dead cell and debris are collected with a short centrifugation and then the media was filtered through a 0.45 µM low protein binding filter. To infect the target cell line, 1 million cells were diluted into 3 ml of cell media and combined with 1 ml of polybrene (final concentration 8 µg/ml) and 1 ml of Lentiviral media in a T25 flask. Once cells reached a confluency of 70%, selection was initiated: media containing 0.5 µg/ml of puromycin (Gibco) was added to the cells and exchanged every 2-3 days. Once cells had recovered and expanded, aliquots were used to validate the restoration of SHLD2 expression by western blot and to perform colony formation assays.

**Chemicals.** Olaparib, carboplatin, staurosporine were purchased from Selleck Chemicals.

**siRNA screen.** A siRNA library was purchased from Dharmacon (1418 siRNAs). Each plate was supplemented with negative siCONTROL (12 wells; Dharmacon) and positive control (four wells, siPLK1, Dharmacon). After optimising high-throughput transfection conditions as described in ref. [61] cells were reverse transfected in a 384 well-plate format with siRNAs. Twenty-four hours after transfection, cells were then exposed to ART558, olaparib or the compound vehicle, DMSO. After 96 hrs compound exposure, cell viability was estimated via CellTiterGlo reagent (Promega). Triplicate screens were performed and the data combined during analysis. Per-well luminescence values were log transformed and then normalized to plate medians (PM) to account for plate-to-plate variation. The extent of increased/decreased drug sensitivity caused by each siRNA (known as Drug Effect or DE scores) were calculated by subtracting normalized values in DMSO-exposed wells from normalized values in small molecule-exposed wells using the equation: $DE = (\log_2$ PM normalized signal of siRNA in the presence of ART558)—$(\log_2$ PM normalized signal of siRNA in the absence of ART558). DE scores were Z standardized according to screen median and median absolute deviation values and are described in Supplementary Data 1.

**siRNA or crRNA transfection.** Lipofectamine RNAimax was used according to the manufacturer's instructions (Invitrogen). Cells were transfected with the indicated siRNAs (Dharmacon) targeting the appropriate gene. Lipofectamine CRISPRmax was used according to the manufacturer's instructions (Invitrogen). Cells were transfected with the indicated Edit-R crRNA (Dharmacon), Edit-R TracrRNA RNA (Dharmacon) and recombinant Cas9. For siRNA and crRNA reagents used, see Supplementary Methods and Supplementary Table 2.

**Viability and clonogenic survival assays.** Cells were seeded in 24- or 6-well plates. Cell viability at the end of the assay was measured using CellTiter-Glo, or colonies stained using sulphorhodamine-B (Sigma), methylene blue or crystal violet and counted. Surviving Fractions (SF) were calculated and drug sensitivity curves plotted as previously described[54]. For full details for each cell line used, see Supplementary Methods.

**Patient-derived organoid (PDO) generation.** Human breast tumour samples were obtained from adult female patients after informed consent as part of a non-interventional clinical trial (BTBC study REC no.: 13/LO/1248, IRAS ID 131133; Principal Investigator: Andrew Tutt; Study Title: 'Analysis of functional immune cell stroma and malignant cell interactions in breast cancer in order to discover and develop diagnostics and therapies in breast cancer subtypes'). This study had local research ethics committee approval and was conducted adhering to the principles of the Declaration of Helsinki. Specimens were collected from surgery and transported immediately to cut up. A clinician histopathologist or pathology-trained technician identified and collected tumour material into basal culture media. Tumour samples were coarsely minced with scalpels and then dissociated using a Gentle MACS dissociator (Miltenyi). The resulting cell suspension was

mechanically disrupted, filtered and centrifuged. Resulting cell pellets were then plated into 3D cultures at $\sim 1$–$2 \times 10^3$ cells/µL in Ocello PDX media (OcellO B.V) and hydrogel as described previously[62–64].

**Ex-vivo 3D viability assays in patient-derived organoid (PDO) models.** Patient-derived organoids from prostate tumours (PR6511 and PR6512) from the CRO repository were screened by Crown BioScience following optimised protocols for 3D viability assay (https://www.crownbio.com/oncology/ex-vivo-services/3d-assays/). Briefly, patient derived xenotransplant were grown in immunodeficient mice, collected and disaggregated, mixed with STO growth supporting cells (ATCC, CRL-1503) 5:1 tumour:STO and resuspended in media that allow their growth in 3D in MW96 according to Crown BioScience Standard Operating Procedures. The day after seeding, media with dilution of the compounds is added to the 3D cell culture to achieve the desired final concentration. The PDO seeded plates were treated with a 9 point curve of ART558, olaparib (S1060, SelleckChem), carboplatin (S1215 SelleckChem) with 3.16 fold dilution each point starting with 50, 30 and 100 µM concentration each, respectively; medium with matched compound concentration was replenished on day 6 and 11. After 14 days of incubation with the compounds, 50 µl of 3D-GloTM (Promega #G9681) reagent is added to each well and luminescence is to be measured using a VarioSkan Flash plate reader. Normalized viability was calculated by normalization of compound treated wells to DMSO treated wells; curves and SF50 were generated with GraphPad Prism.

**In vivo assessment of ART812.** Female SRG OncoRats (SD-$Rag2^{tm2Hera}$ $Il2rg^{tm1Hera}$/HeraArc), were subcutaneously injected with $2 \times 10^7$ MDA-MB-436 $BRCA1$ mutant $SHLD2^{-/-}$ cells in 200 µL of RPMI 1640 containing 50% (v/v) matrigel (ref: 356237, Corning® Matrigel®) into the right flank. Twenty-four days following implantation, rats with mean tumour volume 250–350 mm³, were randomised into two groups. Animals then received daily administration of either vehicle or ART812 (100 mg/kg) for 76 subsequent days. Vehicle was 10% (v/v) DMSO + 10% (v/v) Killiphore® Solutol HS15 (Sigma 42966) made in sterile water. Body weights and tumour volume were measured at least twice a week. Animal housing and experimental procedures were conducted according to the Guide for the care and use of experimental animals issued by the Canadian Council on Animal Care and the National Research Council Guide for the Care and Use of Laboratory Animals.

**Real time cell proliferation and apoptosis assay.** Cells in exponential growth phase were seeded in 100 µL of media at 400 and 1500 cells/well in MW96 plates for DLD-1 parental and DLD-1 BRCA2 KO, respectively. The day after, media was removed and replaced with 90 µL of media containing 5 µM IncuCyte® Caspase-3/7 Green Apoptosis Assay Reagent (Sartorius) and 10 µL of media containing 10× dilution of the compound. Each experimental point was performed in technical triplicate; controls wells to define the analysis setting were included as follows: no cells (media only), cells without IncuCyte® Caspase-3/7 Green Apoptosis Assay Reagent and cells treated with Staurosporine (S1421, SelleckChem) 100 nM as positive control of apoptosis and cell death. DLD-1 parental cells were exposed to vehicle, ART558 5 µM and olaparib 500 nM (S1060, SelleckChem) or combination of ART558 5 µM + olaparib 500 nM; DLD-1 BRCA2$^{-/-}$ cells were exposed to vehicle, ART558 5 µM, olaparib 50 nM or combination of ART558 5 µM + olaparib 50 nM. The plates were then incubated in an IncuCyte® S3 system (v6.2.9200.0, Sartorius) and 4 images/well with 10× objective were acquired every 2 h in bright field (for confluence) and green fluorescence 488 nm wavelength (for apoptosis readout). Analysis settings were optimized to measure the area of confluence within the bright field, and the area of green positivity for apoptosis using positive and negative controls. The proliferation rate was measured as change in % confluence over the time; the % of apoptosis was calculated as a ratio between the green area (apoptosis signal) and total confluence area and plotted over time. Representative pictures displaying overlay of bright field and green fluorescent signal are shown.

***Exo1, Blm* and *Dna2* gene expression.** RNA samples extracted using the RNeasy kit (Qiagen) were reverse transcribed using the Reverse Transcriptase kit (Roche) and oligo dT primers. The resulting cDNA was used to analyze the gene expression of Blm (Mm00476150_m1), Exo1 (Mm00516302_m1) and Dna2 (Mm01169107_m1) using Taqman Gene expression assay. Gene expression values were normalised to expression of the housekeeping gene *Gapdh*.

**Immunoblotting.** Cells were lysed in RIPA lysis and extraction buffer (Thermo Fisher Scientific) supplemented with 1 tablet/10 ml lysis buffer of cOmplete™, EDTA-free Protease Inhibitor Cocktail (Roche) and 1 tablet/10 ml lysis buffer of phosphatase inhibitor cocktail PhosSTOP (Sigma). Lysates were generated on ice, and centrifuged 10 min at $16,900 \times g$ prior to supernatant collection. Supernatants were then subjected to electrophoresis using NuPAGE™ 4–12% (v/v) Bis–Tris or 3–8% Tris-Acetate precast gels (Invitrogen). After migration, proteins were transferred to a nitrocellulose membrane (GE Healthcare). Membranes were blocked using 5% (w/v) milk in TBS buffer supplemented with 0.05% Tween 20 (TBST) at room temperature (RT) for 1 h. Primary antibodies were diluted in 5% (w/v) milk in TBST, and incubated at 4 °C overnight. The next day, the membrane

was washed three times with TBST, each for 10 min, followed by incubation with horseradish-peroxidase-conjugated or fluorophore-conjugated secondary antibodies (LI-COR) at RT for 1 h, in 5% (w/v) milk in TBST. The membrane was washed again three times with TBST, and incubated with Amersham ECL prime detection reagent (GE Healthcare) or imaged using LI-COR Odyssey (ImageStudio v5.2). The membrane was then exposed to X-ray film and the film developed in a darkroom.

For detection of Polθ, cells grown in 6-well plates were washed in PBS, lysed directly in Laemmli buffer (2% SDS, 10% Glycerol, 62.5 mM Tris-HCL pH 6.8) and boiled for 10 min. Samples were syringed five times through a 27G needle. Protein extracts were quantitated using the Bicinchoninic acid assay (ThermoFisher) against a BSA standard curve and made up in 4× NuPAGE LDS sample loading buffer (Invitrogen) supplemented with β-mercaptoethanol. lysates (40 μg) were resolved by SDS-polyacrylamide gel electrophoresis (SDS-PAGE) on NuPAGE 3–8% Tris-Acetate gels (Invitrogen) in NuPAGE Tris-Acetate running buffer (Invitrogen) and wet-transferred in 1× NuPAGE Transfer Buffer (Invitrogen), 20% ethanol and 0.05% SDS to nitrocellulose membranes (Millipore). 5% (w/v) BSA/ Tris-buffered saline + 0.01% (v/v) Tween-20 (TBST) was used for all blocking and incubation steps. Polθ protein was detected by probing the blot overnight at 4 °C with mouse monoclonal anti-Polθ antibody (1:5000, kind gift of Jean-Sébastien Hoffman, CRCT Toulouse) diluted in blocking buffer (5% BSA/TBST). As a loading control, levels of vinculin were determined by probing the membrane with mouse monoclonal anti-vinculin antibody (1:1000, SCBT sc-73614). The membrane was washed thrice for 5 min with TBST and incubated with HRP-conjugated goat anti-mouse IgG (Invitrogen 31430, 1:10,000) for 1 h at room temperature. After five 5 min washes with TBST, signals were detected with ECL detection reagent (GE Healthcare) and imaged on an Amersham Imager 600RGB.

All western blots were repeated independently at least two times with similar results.

For antibody details, see Supplementary Methods.

**Immunofluorescence and image analysis.** For nuclear γH2Ax and RPA foci quantification, cells were seeded in 96-well plates. Cells were fixed in 4% (v/v) paraformaldehyde (PFA) in PBS for 10 min at room temperature (RT), washed twice with PBS, and permeabilized with 0.5% (v/v) Triton X-100 in PBS for 10 min. After two additional washes, cells were blocked with 2% (w/v) BSA, 1% (v/v) FBS in PBS (IFF) for 1 h at RT. Cells were then incubated with primary antibodies in IFF at 4 °C overnight. The cells were then washed three times with PBS, each for 10 min, followed by incubation with Alexa Fluor 555-conjugated mouse and Alexa Fluor 488-conjugated rabbit secondary antibodies (Thermo Fisher Scientific), 1 μg/ ml DAPI in IFF for 1 h at RT. Cells were then washed three times with PBS, and 100 μl PBS was finally added to each well prior to imaging. Plates were imaged using an Image Express high-content imaging system. Quantification of the number of γH2Ax foci and RPA foci (only PCNA-positive cells were used in the analysis) was performed under identical microscopy settings between samples, using the MetaExpress image analysis system (MolDev).

For nuclear pRPA foci quantification cells were seeded on 13 mm coverslips. Cells were fixed in 4% (v/v) paraformaldehyde (PFA) in PBS for 10 min at room temperature (RT), washed twice with PBS, and permeabilized with 0.5% (v/v) Triton X-100 in PBS for 10 min. After two additional washes, cells were blocked with 2% (w/v) BSA, 1% (v/v) FBS in PBS (IFF) for 1 h at RT. Cells were then incubated with primary antibodies in IFF at 4 °C overnight. The cells were then washed three times with PBS, each for 10 min, followed by incubation with Alexa Fluor 488–conjugated rabbit secondary antibodies (Thermo Fisher Scientific) in IFF for 1 h at RT. Cells were then washed three times with PBS, dried and mounted in Vectashield containing DAPI and imaged at ×60 on a Zeiss LSM 780.

**Detection of incorporated BrdU in ssDNA by nondenaturing immuno-fluorescence staining.** To measure levels of ssDNA using a nondenaturing BrdU IF staining procedure, cells cultured on coverslips were first incubated with BrdU (30 μM) for 24 h. As a control, cells were exposed to 2 mM hydroxyurea for 4 h prior to harvesting. Cells were then incubated with extraction buffer (10 mM PIPES pH 6.8, 100 mM NaCl, 300 mM sucrose, 1.5 mM MgCl2 and 0.5% (v/v) Triton X-100) for 2 min on ice. Subsequently, cells were fixed with 4% (v/v) paraf-ormaldehyde in PBS at room temperature for 10 min. After washing with PBS, cells were blocked in 2% (w/v) BSA, 1% (v/v) FBS in PBS (IFF) 1 h at RT. Cells were then incubated for 2 h with anti-BrdU antibody diluted in IFF at room temperature. Subsequently, cells were washed three times with PBS containing 0.05% (v/v) Tween-20 before incubation with secondary antibody. After washing three times with PBS containing 0.05% (v/v) Tween-20, cells were mounted in Vectashield containing DAPI and imaged at ×63 on a Zeiss LSM 780 confocal microscope. The BrdU signal in individual nuclei (defined by the DAPI-stained area) was deter-mined using ImageJ. Images of randomly selected cells for each sample were analysed.

**Mitotic spreads.** Following exposure to the indicated treatment, cells were incu-bated with 0.5% (w/v) colchicine for 4 h. Cells were harvested, washed in PBS and incubated in 0.56% (w/v) KCl at 37 °C for 15 min. Samples were then fixed (3:1 methanol:acetic acid). Cell solutions were dropped onto clean coverslips and

mounted in Vectashield containing DAPI and mitotic spreads imaged at ×60 on a Zeiss LSM 780 confocal microscope.

**Measurement of resection.** ER-AsiSI U2OS cells were reverse-transfected with the mentioned siRNAs and after 24 h exposed to 10μM ART558 or DMSO for an additional 48 h. Cells were trypsinized, centrifuged and resuspended with 37 °C 0.6% low-gelling point agarose (BD Biosciences) in PBS (Gibco) at a concentration of $6 \times 10^6$ cells/ml. A 50-μl cell suspension was dropped on a piece of Parafilm (Pechiney) to generate a solidified agar ball, which was then transferred to a 1.5-ml Eppendorf tube. The agar ball was treated with 1 ml of ESP buffer (0.5 M EDTA, 2% N-lauroylsarcosine, 1 mg/ml proteinase-K, 1 mM CaCl2, pH 8.0) for 20 h at 16 °C while shaking, followed by treatment with 1 ml of HS buffer (1.85 M NaCl, 0.15 M KCl, 5 mM MgCl2, 2 mM EDTA, 4 mM Tris, 0.5% Triton X-100, pH 7.5) for 20 h at 16 °C while shaking. After washing with 1 ml of PBS for 5 × 1 h at 4 °C with rotation, the agar ball was melted by placing the tube in a 70 °C heat block for 10 min. The melted sample was diluted 7-fold with 70 °C ddH2O, mixed with equal volume of appropriate 2× NEB restriction enzyme buffer and stored at 4 °C for future use.

The level of resection adjacent to specific DSBs was measured by quantitative polymerase chain reaction (qPCR) using a modification of the method[43]. The sequences of qPCR primers are shown in Supplementary Table 3. Twenty μL of genomic DNA sample (∼140 ng in 1× NEB restriction enzyme buffer 4) was digested or mock digested with 20 units of restriction enzymes (BsrGI, or HindIII-HF; New England Biolabs) at 37 °C overnight. Two μL of digested or mock-digested samples (∼20 ng) were used as templates in 20 μL of qPCR reaction containing 10 μl of 2× Sybr Green PCR Master Mix (Thermo), 0.5 μM of each primer on an Applied Biosystems® QuantStudio™ 6 Flex. The percentage of ssDNA (ssDNA%) generated by resection at selected sites was determined. Briefly, for each sample, a ΔCt was calculated by subtracting the Ct value of the mock-digested sample from the Ct value of the digested sample. The ssDNA% was calculated with the following equation: '% digested-resistant' = $1/(2^{(\Delta Ct-1)} + 0.5) \times 100$.

**Statistics and reproducibility.** Numbers of independent replicates are included in each figure legend as are details of numbers of events counted.

**Reporting summary.** Further information on research design is available in the Nature Research Reporting Summary linked to this article.

## Data availability

All data and materials used in the analysis are provided within the manuscript. Source data are provided with this paper.

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

## Acknowledgements

We thank Daniel Durocher (Lunenfeld-Tanenbaum Research Institute, Toronto) for providing RPE1 cells, Jean-Sébastien Hoffmann (Cancer Research Center of Toulouse) for providing Polθ antibody, Jeremy Stark (Beckman Research Institute of the City of Hope, Duarte) for providing advice on the use of reporters, Jessica Downs and Wojciech Niedzwiedz (Institute of Cancer Research, London) for provision of the *Asi*SI-ER-U20S cell line, Fredrik Wallberg and Kai Betteridge (Microscopy Facility, Institute of Cancer Research, London) for microscopy support. This work was funded by Cancer Research UK (as part of CRUK Programme Funding C30061/A24439 to S.J.P. and C.J.L.), Breast Cancer Now (as part of BCN Programme Funding to C.J.L./A.T. the Breast Cancer Now Toby Robins Research Centre), NC3Rs funding to A.T. and C.J.L. (NC/P001262/1), Breast Cancer Now, working in partnership with Walk the Walk, for supporting the work of the Breast Cancer Now Centre Patient Derived Models Team, National Institute of Health (as part of Intramural Research Program funding to A.N.) and Artios Pharma. This work represents independent research supported by the National Institute for Health Research (NIHR) Biomedical Research Centre at The Royal Marsden NHS Foundation Trust and the Institute of Cancer Research, London. The views expressed are those of the author(s) and not necessarily those of the NIHR or the Department of Health and Social Care.

## Author contributions

Generated and analysed data and/or developed methodology: D.Z., H.M.R.R., O.A., M.B., H.F., L.G., D.G., V.G., R.A.H., S.L., J.M., C.M.cW., N.M.B.M., S.M., J.N., E.R., M.R., T.S., M.St., K.W., R.B., S.S., A.G., N.S., L.M.B., D.N., E.G.K., R.M., S.H., E.C., G.H., J.S., R.P., C.A., A.F., C.Be., P.B., C.Bo., M.Ca., M.Ch., J.C., T.E., K.E., W.K., E.Mac.D., H.Mc.C., L.P., C.P., L.R., M.Sw., E.W., S.W., A.C.W., A.N., M.T., A.T., S.J.B., G.S.H., S.J.P., G.C.M.S. and C.J.L. Drafting the manuscript: D.Z., H.M.R.R., S.J.P., G.C.M.S. and C.J.L.

Editing and approving the final submitted manuscript: all authors. Conceptualisation of the study: D.Z., H.M.R.R., C.J.L., S.J.P. and G.C.M.S. Supervision and funding acquisition: A.T., S.J.P., C.J.L., G.C.M.S. and N.M.B.M.

## Competing interests

C.J.L. received research funding from: AstraZeneca, Merck KGaA and Artios, received consultancy fees, SAB membership fees or honoraria payments from: Syncona, Sun Pharma, GLG, Merck KGaA, Vertex, AstraZeneca, Tango, 3rd Rock, Ono Pharma, ArtiosPharma Ltd, Abingworth. C.J.L. has stock in: Tango, Ovibio. C.J.L., A.T., S.J.P. and D.Z. are named inventors on patents describing the use of DNA repair inhibitors and stand to gain from their development as part of the ICR 'Rewards to Inventors' scheme. G.S.H. received consultancy fees from, and is a shareholder of ArtiosPharma Ltd. H.M.R.R., O.A., M.B., S.J.B., H.F., L.G., D.G., V.G., R.A.H., S.L., J.M., C.M., N.M.B.M., S.M., J.N., K.P., E.R., M.R., T.S., M.S. and G.C.M.S. are all employees and shareholders of ArtiosPharma Ltd. G.C.M.S. is a shareholder of AstraZeneca PLC. S.J.B. is a Founder Scientist of ArtiosPharma Ltd. All other authors declare no competing interests.
