## [Peer Review File · Nature Communications]

Reviewers' Comments:

Reviewer #1:

Remarks to the Author:

The authors have extensively addressed the concerns raised by this reviewer and the manuscript is greatly improved. All the chemistry issues have been resolved. The authors also present yet another PolTheta inhibitor ART812 that importantly demonstrates in vivo anti-cancer efficacy, which was the main concern raised by this and all other reviewers. Together, this is a comprehensive study with a wealth of information on the Polθ inhibitor ART558. The authors want to publish SAR and a co-crystal structure in a more specialized journal, which is acceptable. While the in vivo data is an important addition, it is also common to use several different in vivo models and PDX models. In this case, this reviewer accepts the arguments raised by the authors that this can come at a later stage. Clearly, there are some complexities on the in vivo models that needs to be sorted out in future studies. Mechanistically, not all issues are addressed, such as PolTheta trapping etc. Again, this reviewer wants to highlight that ART558 is targeting a polymerase, which is highly novel and likely challenging. Altogether, the work presented here is timely, of high quality and will render large interest in a broad scientific community.

Reviewer #2:

Remarks to the Author:

The authors have made substantial changes that have improved the manuscript. In particular, the additional POLQ inhibitor, ART812, reported in the revised version seems effective in vivo against MDA-MB-436 BRCA1/SHL2-deficient tumours, which are PARPi resistant. These in vivo data increase the confidence in the potential of their POLQ inhibitors for further therapeutic development. As most of my concerns were addressed, and figure quality is superior to the previous version, I consider the manuscript suitable for publication.

Reviewer #3:

Remarks to the Author:

The authors have done substantial efforts to answer to the reviewers comments and therefore I recommend publication.

REVIEWERS' COMMENTS

Reviewer #1 (Remarks to the Author):

The authors have extensively addressed the concerns raised by this reviewer and the manuscript is greatly improved. All the chemistry issues have been resolved. The authors also present yet another PolTheta inhibitor ART812 that importantly demonstrates in vivo anti-cancer efficacy, which was the main concern raised by this and all other reviewers. Together, this is a comprehensive study with a wealth of information on the Polθ inhibitor ART558. The authors want to publish SAR and a co-crystal structure in a more specialized journal, which is acceptable.

While the in vivo data is an important addition, it is also common to use several different in vivo models and PDX models. In this case, this reviewer accepts the arguments raised by the authors that this can come at a later stage. Clearly, there are some complexities on the in vivo models that needs to be sorted out in future studies. Mechanistically, not all issues are addressed, such as PolTheta trapping etc. Again, this reviewer wants to highlight that ART558 is targeting a polymerase, which is highly novel and likely challenging. Altogether, the work presented here is timely, of high quality and will render large interest in a broad scientific community.

Our response: We thank the reviewer for taking the considerable time to review our work. We agree that the addition of the *in vivo* experiment provides the first proof-of-concept that a Polθ polymerase inhibitor can elicit anti-tumour efficacy and will be followed up by a more substantial *in vivo* testing programme, to be described in future reports.

End of Referee #1 comments

Reviewer #2 (Remarks to the Author):

The authors have made substantial changes that have improved the manuscript. In particular, the additional POLQ inhibitor, ART812, reported in the revised version seems effective in vivo against MDA-MB-436 BRCA1/SHL2-deficient tumours, which are PARPi resistant. These in vivo data increase the confidence in the potential of their POLQ inhibitors for further therapeutic development. As most of my concerns were addressed, and figure quality is superior to the previous version, I consider the manuscript suitable for publication.

Our response: We thank the reviewer for taking the considerable time to review our work.

End of Referee #2 comments

Reviewer #3 (Remarks to the Author):

The authors have done substantial efforts to answer to the reviewers comments and therefore I recommend publication.

Our response: We thank the reviewer for taking the considerable time to review our work.

End of Referee #3 comments